# Interface engineering enabling thin lithium metal electrodes down to 0.78 μm for garnet-type solid-state batteries

Weijie Ji [1], Bi Luo[1], Qi Wang[1], Guihui Yu[1], Zixun Zhang[1], Yi Tian[1], Zaowen Zhao[2], Ruirui Zhao [3], Shubin Wang[4], Xiaowei Wang [1] ✉, Bao Zhang[1], Jiafeng Zhang[1] ✉, Zhiyuan Sang[5] & Ji Liang [6] ✉

Controllable engineering of thin lithium (Li) metal is essential for increasing the energy density of solid-state batteries and clarifying the interfacial evolution mechanisms of a lithium metal negative electrode. However, fabricating a thin lithium electrode faces significant challenges due to the fragility and high viscosity of Li metal. Herein, through facile treatment of Ta-doped $Li_7La_3Zr_2O_{12}$ (LLZTO) with trifluoromethanesulfonic acid, its surface $Li_2CO_3$ species is converted into a lithiophilic layer with $LiCF_3SO_3$ and LiF components. It enables the thickness control of Li metal negative electrodes, ranging from 0.78 μm to 30 μm. Quasi-solid-state lithium-metal battery with an optimized 7.54 μm-thick lithium metal negative electrode, a commercial $LiNi_{0.83}Co_{0.11}Mn_{0.06}O_2$ positive electrode, and a negative/positive electrode capacity ratio of 1.1 shows a 500 cycles lifespan with a final discharge specific capacity of 99 mAh $g^{-1}$ at 2.35 mA $cm^{-2}$ and 25 °C. Through multi-scale characterizations of the thin lithium negative electrode, we clarify the multi-dimensional compositional evolution and failure mechanisms of lithium-deficient and -rich regions (0.78 μm and 7.54 μm), on its surface, inside it, or at the Li/LLZTO interface.

With the rapid advancement of portable electronic devices and new energy vehicles, an increasingly rigid requirement has now been imposed on lithium-ion batteries (LIBs) in terms of energy density and safety[1–3]. However, current commercial LIBs with liquid electrolytes confront intrinsic constraints that originate from their structures, fundamentally impeding them from meeting these requirements[4,5]. Consequently, solid-state batteries have been proposed and rapidly attracted substantial attention due to their high safety features, including non-flammability and non-leakage[6]. Besides, the typical

solid-state electrolytes (SSEs), such as $Li_7La_3Zr_2O_{12}$ (LLZO), also exhibit higher compatibility toward lithium (Li) and can be directly coupled with Li metal negative electrodes, which offers higher energy density than current LIBs[7,8]. Therefore, solid-state lithium-metal batteries (SSLMBs) stand as a state-of-the-art candidate for the next generation high-energy-density and high-safety rechargeable batteries.

However, the practical application of SSLMBs confronts a series of significant challenges, primarily associated with the fabrication of lithium metal negative electrodes. Specifically, the inherent high

[1]National Engineering Laboratory for High-Efficiency Recovery of Refractory Nonferrous Metals, School of Metallurgy and Environment, Central South University, Changsha, China. [2]Special Glass Key Lab of Hainan Province, School of Materials Science and Engineering, Hainan University, Haikou, China. [3]School of Chemistry, Engineering Research Center of MTEES (Ministry of Education), South China Normal University, Guangzhou, Guangdong, China. [4]State Environmental Protection Key Laboratory of Urban Ecological Environment Simulation and Protection, South China Institute of Environmental Sciences, Ministry of Ecology and Environment (MEE), Guangzhou, China. [5]School of Materials Science & Engineering, Peking University, Beijing, China. [6]Key Laboratory for Advanced Ceramics and Machining Technology of Ministry of Education, School of Materials Science and Engineering, Tianjin University, Tianjin, China. ✉e-mail: yjywxw@csu.edu.cn; yjyzjf@csu.edu.cn; liangji@tju.edu.cn

viscosity and poor machinability of lithium metal impedes the precise control of its thickness by conventional mechanical compressing technologies[9]. This results in a lithium metal negative electrode, used in both laboratory or industry scenarios, typically with a thickness of several tens to even hundreds of micrometers, which not only leads to the wastage of this costly metal resource but also significantly compromises the energy density of SSLMBs[10]. Furthermore, owing to the high reactivity of lithium metal, batteries containing an excessive amount of it are susceptible to combustion or even explosion in the event of battery failure or accidents, thereby presenting substantial safety hazards[11].

In addition, a Li metal negative electrode with an excessive thickness also hinders the in-depth investigation of the intrinsic operation/failure mechanism of SSLMBs. Specifically, current characterization techniques encounter difficulties in probing inside lithium metals, limiting their ability to explore the multi-dimensional composition/structural evolution over lithium metal surfaces, inside their bulk phase, or even at the Li|SSE interfaces. Consequently, the differentiation and quantification of various forms of lithium species (e.g., the active and inactive lithium) inside lithium metal negative electrodes, as well as their distribution and evolution during continuous battery operation, remains notably challenging, which is, however, essential for understanding the operation/failure mechanism of SSLMBs[12]. Regarding this, the controllable thinning of lithium metal would greatly facilitate the fundamental study of lithium-metal batteries (LMBs) as well.

Consequently, the controllable construction of thin lithium metal negative electrodes would be critical for improving battery energy density and safety and, more importantly, for fully and accurately exploring battery operation/failure mechanisms. By far, significant efforts have been exerted for fabricating thin lithium metal, such as electrochemical deposition[13], vacuum evaporation[14], mechanical rolling[9], and anodic compositing[15]. Unfortunately, the lithium metal negative electrodes obtained by these technologies are expensive and often fairly complicated, and the controllable fabrication of thin lithium metal negative electrodes thinner than 10 μm with an acceptable cost-effectiveness is still extremely hard to achieve[16,17]. In addition, these thin lithium negative electrodes are mainly adopted in liquid lithium battery systems, and the severe side reactions between liquid electrolytes and lithium metal cause rapid consumption of lithium metal and the formation of dead lithium; substantially compromising the battery performance, especially regarding the cycling life[15,18]. Although these side reactions would be largely suppressed by simple interface modification of the SSEs[18], these free-standing lithium metal negative electrodes still cannot be directly adopted in SSLMBs due to their intrinsically poor contact with typical SSEs. Therefore, the controllable and direct fabrication of thin lithium metal over SSLMBs is not only an essential factor for enhancing the performance SSLMBs in terms of their safety, energy density, and cycling lifespan but also highly desirable for fundamentally understanding their operation/failure mechanism by achieving detailed full dimensional characterization of lithium metal. However, such controllable and direct thinning of lithium metal over SSEs has not yet been achieved by far.

On the basis of these considerations, we herein report a very facile yet effective strategy for precisely tuning the thickness of thin lithium metal (0.78–30 μm) over SSEs and significantly improving the cycling stability of thin LMBs. In this strategy, the direct coating of a thin lithium metal over LLZO was achieved by via constructing a super lithiophilic layer on LLZO's surface. On the one hand, this strategy successfully enhances the utilization of lithium metal and improves the energy density and cycling stability of SSLMBs. A symmetric Li ||Li cell can stably operate for up to 800 h at 1.0 mA cm$^{-2}$. Furthermore, batteries with a negative/positive capacity ratio (N/P ratio) of 1.1, equipped with thin lithium metal negative electrodes with a thickness of 7.54 μm and LiNi$_{0.83}$Co$_{0.11}$Mn$_{0.06}$O$_2$ (NCM) positive electrodes, exhibit a remarkably stable operation over 500 cycles at a high current density of 2.35 mA cm$^{-2}$. On the other hand, more importantly, the precise control of thin lithium metal also enables the straightforward exploration of the compositional and structural evolution inside the bulk phase of lithium metal negative electrodes, clarifying the functioning and failure mechanisms of both lithium-deficient and lithium-rich regions during battery operation. Therefore, this work provides an effective strategy for constructing thin lithium metal negative electrodes, fundamentally clarifies their operational characteristics, and promotes the practical application of SSLMBs.

## Results

### Fabrication and design rationales of the TfOH-modified layer

Considering the high chemical stability[19] and ionic conductivity[20] ($7.0 \times 10^{-4}$ S cm$^{-1}$ at 25 °C) of the Ta-doped Li$_7$La$_3$Zr$_2$O$_{12}$ (LLZTO) material (Supplementary Fig. S1), it was chosen as the SSE in this work. LLZTO was prepared through a high-temperature solid-state sintering process. Specifically, the raw materials are uniformly mixed through ball milling, followed by the first solid-state sintering to generate LLZTO powder. The powder was then cold-pressed and formed, followed by a second densification sintering to obtain dense LLZTO ceramic pellets. LLZTO ceramic pellets need to be polished and cleaned before use and then stored in a glove box filled with Ar.

Due to the fact that almost all the materials processing, except for cold pressing, was carried out in an ambient environment, LLZTO would be inevitably exposed to the air and react with CO$_2$ and H$_2$O, resulting in the formation of a LiOH surface layer through the Li$^+$/H$^+$ exchange, which then combines with CO$_2$ to finally form Li$_2$CO$_3$ on the surface of LLZTO[21]. The poor affinity of Li$_2$CO$_3$ toward lithium metal hinders the effective wetting of LLZTO surface by molten lithium, resulting in significant difficulty in precisely controlling the thickness of the coated Li negative electrode as well as a high interfacial impedance. Meanwhile, the low ionic conductivity of Li$_2$CO$_3$ is also not conducive to lithium ion transport. Therefore, we adopted CF$_3$SO$_3$H (TfOH) solution to in situ transform the Li$_2$CO$_3$ layer on the surface of LLZTO into LiCF$_3$SO$_3$ with a high lithium affinity and thermal stability[22], to improve the interface contact of Li|LLZTO. In addition, we investigated the composition and microstructure of the Li$_2$CO$_3$ layer over LLZTO surface after being exposed to air for 1, 12, and 24 h, along with acid treatment with TfOH solution, respectively. As shown in Supplementary Fig. S2, Li$_2$CO$_3$ were observed on the surface of LLZTO after exposure to air, but Li$_2$CO$_3$ no longer existed after the TfOH treatment. It indicates that TfOH-treatment can completely convert the surface Li$_2$CO$_3$ layer into the desired lithiophilic layer. Moreover, Supplementary Fig. S3 shows the surface microstructure of LLZTO under different exposure times in the air, revealing an increase in Li$_2$CO$_3$ coverage with prolonged exposure time. Notably, after 24 h of air exposure, the LLZTO surface was fully covered with Li$_2$CO$_3$, forming a continuous Li$_2$CO$_3$ layer that is conducive to the subsequent formation of a continuous TfOH-modified layer following acid treatment. As anticipated, significant differences in surface microstructure were observed among TfOH-treated samples subjected to different exposure durations, as evidenced in Supplementary Fig. S4. Specifically, compared to specimens exposed for 1 h and 12 h, LLZTO surfaces exposed for 24 h displayed the formation of a continuous TfOH-modified layer following acid treatment with TfOH solution.

Therefore, in order to ensure material consistency and form a continuous Li$_2$CO$_3$ interlayer to ensure the continuity of the generated TfOH-modified layer, we intentionally exposed LLZTO to the air for 24 h, denoted as air-LLZTO. Specifically, TfOH was dissolved in dimethyl sulfoxide (DMSO) to generate a low-concentration TfOH solution (1 wt%) to protect the LLZTO from excessive reactions. This solution was then dripped onto air-LLZTO and kept for 5 min to completely convert the Li$_2$CO$_3$ layer on the air-LLZTO surface into a TfOH-modified layer. Consequently, we obtained LLZTO with a continuous

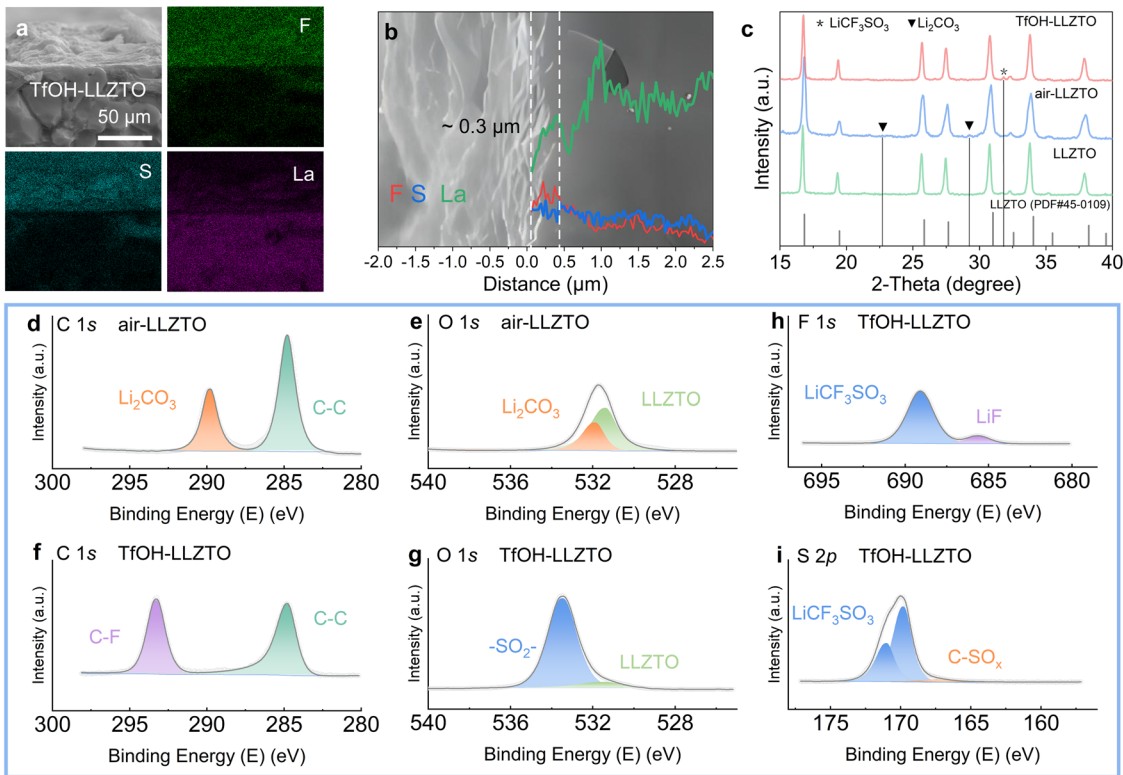

**Fig. 1 | Characterization of the TfOH-modified layer on LLZTO. a** Cross-sectional SEM and EDS images of TfOH-LLZTO. **b** Elemental distributions in the line scan mode of EDS, including the signals of F, S, and La elements. **c** XRD profile of pristine LLZTO, air-LLZTO, and TfOH-LLZTO. **d, e** XPS spectra of air-LLZTO of C 1s and O 1s. **f, g** XPS spectra of TfOH-LLZTO of C 1s and O 1s. **h, i** XPS spectra of TfOH-LLZTO of F 1s and S 2p. Source data for Fig. 1b–i are provided as a Source Data file.

TfOH-modified layer on the surface, i.e., TfOH-LLZTO. This in situ formation of TfOH-modified and lithiophilic interlayer is the prerequisite for the subsequent controllable construction of thin lithium metal on LLZTO. Subsequently, in an argon atmosphere, molten Li was dripped onto the surface of TfOH-LLZTO at 300 °C. Due to the enhanced wettability of the TfOH-modified layer, molten lithium can be easily dispersed on the surface of TfOH-LLZTO using a scraper. By controlling the amount of dripping and scraping, the thickness of lithium metal can be accurately controlled. More details about materials fabrication can be found in the supplementary information.

## Characterization of the TfOH-modified layer

Scanning electron microscopy (SEM) images of TfOH-LLZTO reveal the uniform coverage of a modification layer on the LLZTO surface (Supplementary Fig. S4g). The corresponding elemental mappings obtained through energy dispersive spectroscopy (EDS) reveal homogeneously distributed F and S elements with high intensity (Supplementary Fig. S4h, i), evidencing the successful formation of F- and S-containing modification layer on the surface of TfOH-LLZTO. Besides, the cross-sectional SEM image of a TfOH-LLZTO pellet further verifies the uniformity of this modification layer (Fig. 1a). These findings confirm the effective and uniform in situ modification of the SSE surface via our facile TfOH treatment strategy.

In the high-magnification cross-sectional SEM image of TfOH-LLZTO, it can be clearly seen that the newly formed modification layer is in intimate contact with the LLZTO surface without gaps or voids (Fig. 1b). The corresponding elemental linear scan reveals the boundary between the modification layer and the LLZTO beneath it. Specifically, signals associated with F and S are detected at the surface, i.e., the modification layer region, where the signals corresponding to La is absent. This observation also indicates that the modification layer is approximately 0.3 µm thick. On the basis of these results, the TfOH

treatment of air-LLZTO induces a thin, densely-coated, and evenly distributed surface modification layer over LLZTO, primarily composed of compounds containing F and S elements (Supplementary Table S1). Such homogenous and intimate contact between the surface modification layer and LLZTO beneath it should be favorable for the uniform deposition/stripping of Li, avoiding current focusing and lithium dendrite growth[23].

To further investigate the surface components of TfOH-LLZTO, X-ray diffraction (XRD) was carried out (Fig. 1c). A minor peak located at around 23° and 29°, corresponding to $Li_2CO_3$, appears on air-LLZTO after 24 h exposure to the air. This peak disappears in TfOH-LLZTO, indicating the successful conversion of $Li_2CO_3$ upon TfOH treatment. However, the major characteristic peak intensity of $LiCF_3SO_3$, the expected component in the modification layer, at 31.8°, is lower than those of LLZTO, making them less distinct. Thus, more information about the materials was then obtained through the Raman spectrum and attenuated total reflectance Fourier-transform infrared spectrum (ATR-FTIR). As shown in the Raman spectra (Supplementary Fig. S5a), the minor peaks at 159 and 1091 $cm^{-1}$ represent the vibration of $CO_3^{2-}$, which only exists in air-LLZTO, proving additional evidence of the conversion of $Li_2CO_3$ upon TfOH treatment. ATR-FTIR, a special mode of infrared typically for investigating solid-state and non-transparent materials (Supplementary Fig. S5b), also supports this result. The representative bands of $CO_3^{2-}$ at 858 $cm^{-1}$ only exist in air-LLZTO and are absent in TfOH-LLZTO samples. Particularly, the characteristic bands at 1266 $cm^{-1}$ ($\theta_{as} SO_3$), 1033 $cm^{-1}$ ($\theta_s SO_3$), 1230 $cm^{-1}$ ($\theta_{as} CF_3$), and 1182 $cm^{-1}$($\theta_s CF_3$) for TfOH-LLZTO demonstrates that $LiCF_3SO_3$ has been successfully formed after the TfOH treatment[24], which is also in agreement with the XRD results showing the major characteristic peaks of $LiCF_3SO_3$.

X-ray photoelectron spectroscopy (XPS) was then conducted to gain insights into the surface chemical information of air-LLZTO and

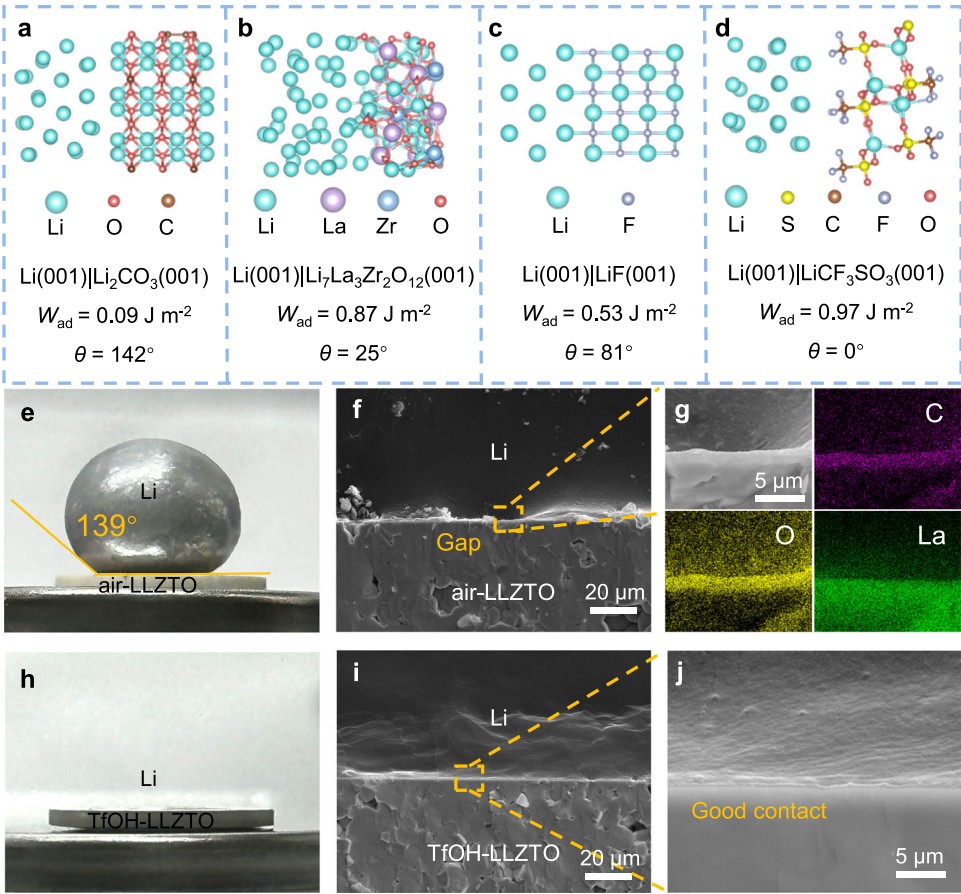

**Fig. 2 | Calculation and characterization of the interfacial wettability of TfOH-modified layer on LLZTO. a–d** Interfacial structure, work of adhesion ($W_{ad}$) of the Li|Li$_2$CO$_3$, Li|Li$_7$La$_3$Zr$_2$O$_{12}$, Li|LiF, and Li|LiCF$_3$SO$_3$ interface. **e** Digital photo of molten Li on the contaminated surface of air-LLZTO. **f** Cross-sectional SEM image of Li|air-TfOH-LLZTO. For air-LLZTO, the C 1*s* peak at 290 eV (Fig. 1d) and O 1*s* signal at 531.8 eV (Fig. 1e) stem from the surface Li$_2$CO$_3$ passivation layer after exposure to the air. After TfOH treatment, these characteristic peaks associated with Li$_2$CO$_3$ are sharply attenuated due to the chemical reaction between Li$_2$CO$_3$ and TfOH (Fig. 1f). For TfOH-LLZTO, the O 1*s* peaks at 533.5 eV correspond to the reaction products of Li$_2$CO$_3$ and TfOH (Fig. 1g)[25]. This TfOH treatment thus results in the complete elimination of surficial Li$_2$CO$_3$. Besides, the F 1*s* signals (685.7 eV and 689 eV) also confirm the generation of LiCF$_3$SO$_3$ on the TfOH-LLZTO surface, together with a small amount of LiF, which should be resulted from minor byproducts of the impurity-induced decomposition of LiCF$_3$SO$_3$ (Fig. 1hand Supplementary Equation S1)[26]. For the S 2*p* spectra (Fig. 1i), the prominent peaks located at 169.88 eV and 171.08 eV again correspond to the formation of LiCF$_3$SO$_3$[27], and the weak peaks at 167.33 eV and 168.63 eV can be assigned to C-SO$_x$ species (Supplementary Equation S2)[28], possibly correlated with the defluorination of CF$_3$SO$_3^-$ to produce LiF, as detailed in Supplementary Note S1. The approach of surface chemical engineering thus effectively transforms surface Li$_2$CO$_3$ layer on LLZTO into LiCF$_3$SO$_3$ and LiF (Supplementary Fig. S6).

## Optimized interfacial lithiophility

The complete conversion of Li$_2$CO$_3$ into a surface modification layer containing LiCF$_3$SO$_3$ with high ionic conductivity and LiF with low electronic conductivity (~ 10$^{-10}$ S cm$^{-1}$) is expected to be beneficial for enhancing the lithiophility and reducing interface impedance[29]. To prove the high surface lithiophilicity of TfOH-LLZTO, density functional theory simulations were first performed. The interfacial models

LLZTO interface with a significant gap. **g** SEM and EDS images of Li|air-LLZTO interface. **h** Digital photo of molten Li on the modified surface of TfOH-LLZTO. **i, j** Cross-sectional SEM images of Li|TfOH-LLZTO interface with intimate contact between Li and modified LLZTO.

of Li|Li$_2$CO$_3$, Li|LLZO, Li|LiCF$_3$SO$_3$, and Li|LiF were constructed using the low-energy surface of Li (001), Li$_2$CO$_3$ (001), LLZO (001), LiCF$_3$SO$_3$ (001), and LiF (001) slabs. As shown, the work of adhesion ($W_{ad}$) for Li|Li$_2$CO$_3$ is 0.09 J m$^{-2}$, which is lower than Li|LLZO (Fig. 2a, Supplementary Data 1). It corresponds to a large contact angle ($\theta$) of 142°, which is much larger than that of Li|LLZO (25°, Fig. 2b, Supplementary Data 2), indicating the poor lithiophilicity of Li$_2$CO$_3$ surface that tends to result in more significant interfacial impedance and inhomogeneous Li deposition/stripping. In comparison, the interfacial adhesion work of Li|LiF is 0.53 J m$^{-2}$ (Fig. 2c, Supplementary Data 3), and the adhesion work of Li|LiCF$_3$SO$_3$ is as high as 0.97 J m$^{-2}$. The calculated contact angle reaches 0° for Li|LiCF$_3$SO$_3$, indicating that Li can be completely wetted on LiCF$_3$SO$_3$ surface (Fig. 2d, Supplementary Data 4). This indicates that the TfOH-modified layer has significantly enhanced lithium affinity, which can make the molten lithium metal evenly spread on its surface, thus of great significance for achieving the thinning of lithium metal as well as intimate interfacial contact. Besides, the existence of LiF with low electronic conductivity is also beneficial for promoting the uniform deposition of lithium and suppressing the growth of lithium dendrites[29].

To verify this theoretical prediction, molten Li was coated on the air-LLZTO and TfOH-LLZTO at 300 °C. As shown in Fig. 2e, the contact angle of Li|air-LLZTO reached 139°, which is consistent with theoretical calculations (Fig. 2a), confirming the high lithium hydrophobicity of the Li$_2$CO$_3$ layer. Consequently, obvious gaps and defects can be found at the interface of Li|air-LLZTO (Fig. 2f), which would lead to a high interface impedance. Figure 2g shows the spatial distribution of C, O, and La elements at the Li|air-LLZTO interface, clearly demonstrating

that a strong enrichment of C at the interface and indicating that the presence of lithiophobic $Li_2CO_3$ is the main factor for the poor the wettability of air-LLZTO for Li. In contrast, the Li|TfOH-LLZTO interface has a very small wetting angle (Fig. 2h), demonstrating its enhanced lithiophilicity, allowing molten lithium metal to freely spread on it. In this case, even a small amount of lithium liquid can uniformly cover the surface of LLZTO, significantly facilitating the subsequent lithium metal scraping and thinning. In addition, an intimate and compact interfacial contact between Li and TfOH-LLZTO was achieved, which is free from gaps or other interfacial defects (Fig. 2i, j), again confirming the significant contributions made by the TfOH-modified layer to the intimate contact between Li and LLZTO.

Thus, theoretical simulations and experimental characterization clearly and collaboratively confirm that the TfOH-modified layer containing $LiCF_3SO_3$ and LiF can significantly improve the interface contact between LLZTO and lithium metal, thus improving the interfacial wettability. This not only effectively reduces interface impedance and promotes efficient lithium ion transport, but also significantly facilitates the subsequent lithium metal scraping and thinning. At the same time, the electronically insulating LiF can inhibit the growth of lithium dendrites, which is beneficial for improving interface stability and extending battery cycle life.

## Electrochemical performance of Li∥Li cells protected by a TfOH-modified layer

The impact of TfOH treatment on the battery performance of TfOH-LLZTO was then assessed in a symmetric cell by galvanostatic cycling combined with electrochemical impedance spectroscopy (EIS). As shown in the corresponding Nyquist plot (Supplementary Fig. S7), the first semicircle represents the bulk impedance ($R_B$), the second semicircle represents the grain boundary impedance ($R_{GB}$), and the third semicircle represents interfacial impedance ($R_{int}$). Prompted by the improved wettability, a significant reduction in $R_{int}$ between Li metal and TfOH-LLZTO is observed in comparison with the Li|air-LLZTO|Li one. It indicates that the TfOH-modification layer effectively improves the interface contact and promotes lithium ion transport.

The initial six cycles of activation for Li|air-LLZTO|Li and Li|TfOH-LLZTO|Li symmetric cells were conducted at low current density ($0.3\,mA\,cm^{-2}$, Fig. 3a and Supplementary Fig. S8a). As shown, the Li|TfOH-LLZTO|Li cell did not exhibit significant polarization changes during these cycles (Fig. 3a). In contrast, the Li|air-LLZTO|Li cell exhibited a continuous increase in polarization and failed in the second cycle (Supplementary Fig. S8a). To further explore the electrochemical behavior of these batteries, relaxation time distribution (DRT) analysis was used, where impedance data was converted from the frequency domain to the time domain, allowing the identification of peaks related to characteristic time constants[30,31]. For the DRT analysis, a peak with a relaxation time less than $10^{-7}\,s$ is related to the ion conduction within the crystal lattice of the LLZTO grain; a peak with a relaxation time of $10^{-7}$ to $10^{-4}\,s$ is related to the ion conductivity at the grain boundary; a peak with a relaxation time of $10^{-3}\,s$ is related to the charge transfer kinetics of the Li negative electrode[32,33]; while a peak with a relaxation time greater than $10^{-3}\,s$ can be attributed to the interface ion transport of Li|LLZTO[34,35]. As shown in the DRT analysis of Li|TfOH-LLZTO|Li symmetric cell Fig. 3b, its impedance was small and constant throughout the cycling process. As for the Li|air-LLZTO|Li counterpart, however, it demonstrated a sudden rise in impedance followed by a micro short circuit, mainly caused by the low charge transfer dynamics at the Li|air-LLZTO interface (Supplementary Fig. S8b).

To enhance the understanding of the temporal evolution of interface resistance during continuous cycling, unidirectional galvanostatic electrochemical impedance spectroscopy (GEIS) was performed on the Li∥Li cells at a current density of $0.5\,mA\,cm^{-2}$ (Fig. 3c). The Li|TfOH-LLZTO|Li cell operated stably at $0.5\,mA\,cm^{-2}$ for over 30 h

(Fig. 3c), without significant change in impedance, and the Li|TfOH-LLZTO interface remains stable (Fig. 3d). This enhanced performance can be attributed to the favorable compositional feature of TfOH-LLZTO surface that induced good interfacial contact between Li|TfOH-LLZTO and inhibited lithium dendrite growth, thus significantly improving the stability. In comparison, the Li|air-LLZTO|Li cell exhibited severe polarization after 0.5 h (Supplementary Fig. S8c), mainly attributed to the stress concentration at grain boundaries caused by the rapid growth of lithium dendrites under high current density, resulting in grain boundary failure (Supplementary Fig. S8d). It can be attributed to the poor lithiophilicity of $Li_2CO_3$ on air-LLZTO surface, leading to lithium dendrites growth at the interface voids and ultimately to battery failure.

Electrochemical kinetics of the lithium stripping/plating of the cells were further evaluated through Tafel analysis (Fig. 3e). In the strongly polarized region, a linear relationship exists between logarithmic current ($\ln i$) and polarization $\eta$, when the voltage deviates from the equilibrium state. Therefore, we have replotted the Tafel results in the form of an Allen Pickling graph (Fig. 3f and Supplementary Fig. S9), and the data processing details are available in the supporting information. As shown in Fig. 3f, the exchange current densities ($i_0$) of Li|TfOH-LLZTO|Li and Li|air-LLZTO|Li are 23.1039 mA and 0.0015 mA, respectively, with extremely significant differences. Considering the high $i_0$ value (23.1039 mA) of Li|TfOH-LLZTO|Li, it shows a clear dynamic advantage compared with Li|air-LLZTO|Li, further demonstrating the effectiveness of the ion transport channel constructed by the TFOH modified layer at the interface. Through further data processing of the Tafel curve, relevant data on charge transfer resistance ($R_{CT}$) was obtained, and the details of data processing are provided in the supporting information. The $R_{CT}$ of Li|TfOH-LLZTO|Li and Li|air-LLZTO|Li are $0.0011\,\Omega$ and $17.1472\,\Omega$, respectively. The extremely low $R_{CT}$ of Li|TfOH-LLZTO|Li thus clearly demonstrates its facile kinetics for lithium stripping/plating, attributed to the rapid lithium ion transport ability of the TfOH-modified interface layer.

In addition, the kinetic performance of the solid electrolyte interface can be described by the critical current density (CCD), which reflects the ability of the interface to resist lithium dendrite penetration under high current densities. As shown in Supplementary Fig. S10, Li|air-LLZTO|Li cell suffered a short-circuit only at $0.2\,mA\,cm^{-2}$, indicating that the limited contact and high interface impedance brought about by the gaps/voids at the Li|air-LLZTO interface hinders the transport of lithium ions, leads to rapid growth lithium dendrites, and results in the penetration of the LLZTO SSE. On the contrary, the Li|TfOH-LLZTO|Li cell can withstand a much higher current density of $2.4\,mA\,cm^{-2}$ without failure at a constant capacity of $0.2\,mAh\,cm^{-2}$ (Fig. 3g). The high CCD of Li|TfOH-LLZTO|Li indicates that the enhanced wettability of the TfOH-modified layer brings about close contact at the interface, which is conducive to the efficient transition of lithium ions and clearly demonstrates its significant ability to resist lithium dendrite growth.

In the long-term cycling test conducted at $0.2\,mA\,cm^{-2}$, $0.2\,mAh\,cm^{-2}$, and $25\,°C$, the Li|TfOH-LLZTO|Li cell again displayed a high stability for over 2500 h with an extremely low overpotential down to only 10 mV (Fig. 3h). Even at a higher current density of $1\,mA\,cm^{-2}$, it can still stably cycle for over 800 h with a small overpotential of ~ 40 mV (Fig. 3i). On the contrary, the Li|air-LLZTO|Li cell displayed a highly fluctuating overpotential of over 50 mV under $0.2\,mA\,cm^{-2}$. The impressive cycling stability of Li|TfOH-LLZTO|Li, benefiting from stable and intimate Li|TfOH-LLZTO interface, thus provides credible evidence of its capability to inhibit Li dendrite formation. In addition, the rapid Li-conducting features of the $LiCF_3SO_3$ layer in TfOH-LLZTO also enable the sufficient Li-ion flux between Li metal and LLZTO electrolyte, thus endowing it with a low overpotential for Li plating/stripping as well[36]. To our knowledge, this good

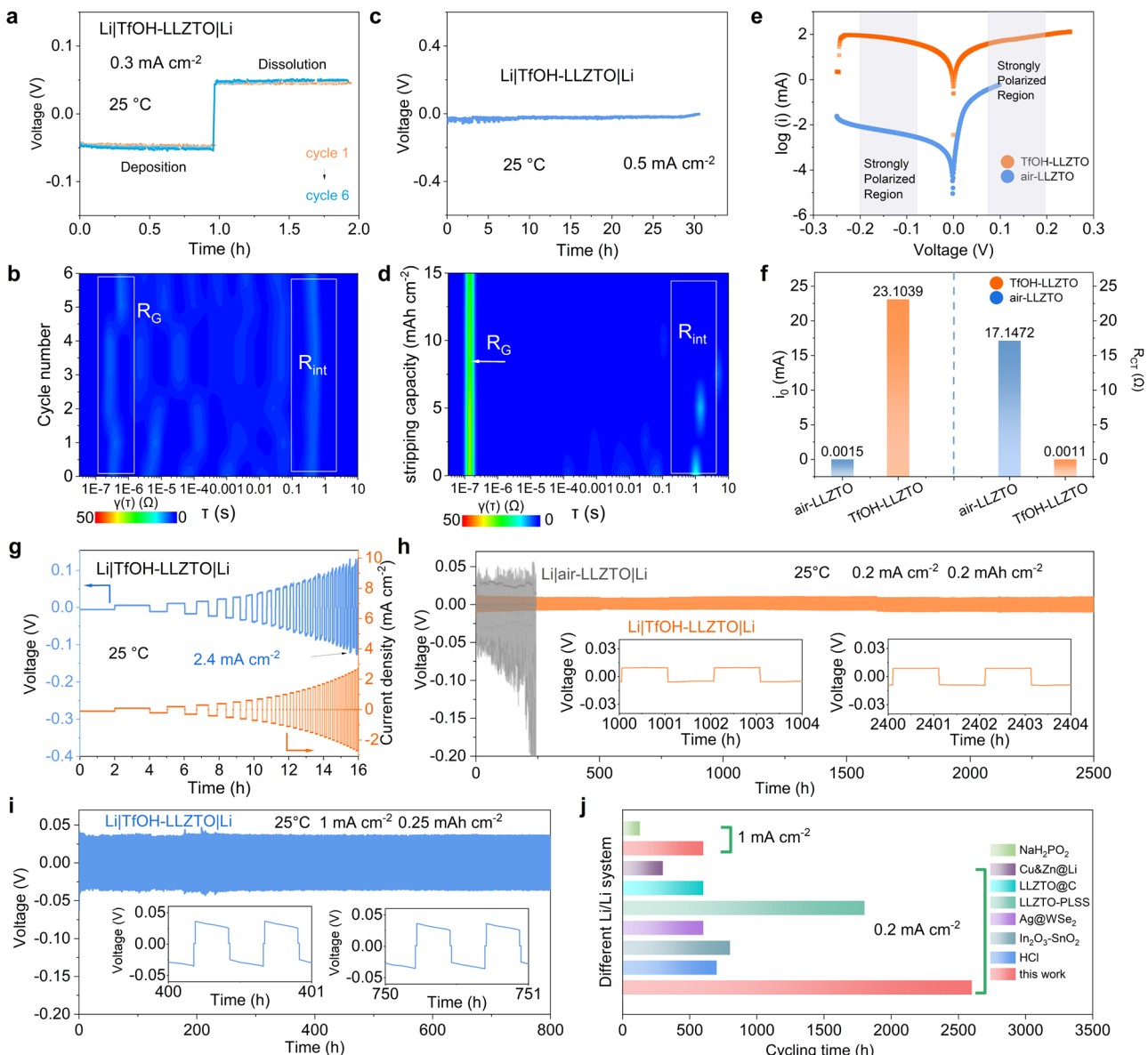

**Fig. 3 | Electrochemical characterizations of Li‖Li symmetrical batteries.**
**a** Voltage profile during dissolution and deposition cycling experiments of a Li|
TfOH-LLZTO|Li cell at 0.3 mA cm$^{-2}$. **b** The DRT transition result of EIS is recorded in
Figure a at the current density of 0.3 mA cm$^{-2}$. Seven sets of data were selected
from the original data for plotting. **c, d** Evolution of polarization voltage and DRT
transition result of EIS recorded during unidirectional charging in a Li|TfOH-
LLZTO|Li cell at 0.5 mA cm$^{-2}$. Seven sets of data were selected from the original
data for plotting in Figure (**d**). **e** Tafel plots of Li‖Li cells. **f** Comparison of exchange
currents ($i_0$) and charge transfer resistance ($R_{CT}$) with Li|air-LLZTO and Li|TfOH-

LLZTO. **g** Voltage-time profiles of Li|TfOH-LLZTO|Li on galvanostatic cycling with a
stepped current density and constant capacity at 25 °C. Yellow represents the
current curve, and blue is the voltage curve. **h** Cycling performance of Li|air-LLZTO|
Li and Li|TfOH-LLZTO|Li cells at 0.2 mA cm$^{-2}$. **i** Cycling performance of a Li|TfOH-
LLZTO|Li cell at 1 mA cm$^{-2}$. **j** Comparison of cycling time for Li‖Li symmetric cells
using TfOH-LLZTO against other reported modification strategies employing
garnet electrolytes at current densities of 0.2 and 1 mA cm$^{-2}$. Source data for Fig. 3
are provided as a Source Data file.

performance of TfOH-LLZTO is better than most of the previous
reports on LLZTO-based SSEs with various surface modification layers
(Fig. 3j)[37–45].

## Precise control of the thickness of Li metal negative electrodes via TfOH-modified layer

Benefiting from the strong affinity of TfOH-LLZTO for lithium metal,
precise thickness regulation of lithium negative electrode is achievable
using a combined quantitative lithium melt-scraping strategy (Fig. 4a).
The surface modification layer comprising LiCF$_3$SO$_3$ and LiF on top of
TfOH-LLZTO allows molten lithium metal to seamlessly disperse
across its surface. Consequently, by controlling the quantity of applied
molten lithium metal, precise regulation of the thickness of lithium

metal negative electrodes can be achieved, resulting in the creation of
thin lithium metal negative electrodes less than 1 µm thick. As shown in
Fig. 4b–g, thin lithium layers with different thicknesses were success-
fully fabricated on the surface of TfOH-LLZTO. The thickness assess-
ment and corresponding standard deviation are depicted in
Supplementary Fig. S11. In addition, to further confirm the consistency
of this technology for fabricating thin lithium layers, a larger piece of
LLZTO (1.43 cm in diameter) was adopted and coated with thin lithium
metal layers of 0.78 µm and 7.54 µm thick, as depicted in Supple-
mentary Fig. S12a, b. The morphology characteristics of the lithium
metal layer from cross-view SEM images show a similar morphology to
that of the smaller size (1.2 cm in diameter). In the top-view SEM ima-
ges, the surfaces of these thin lithium layers are smooth and uniform

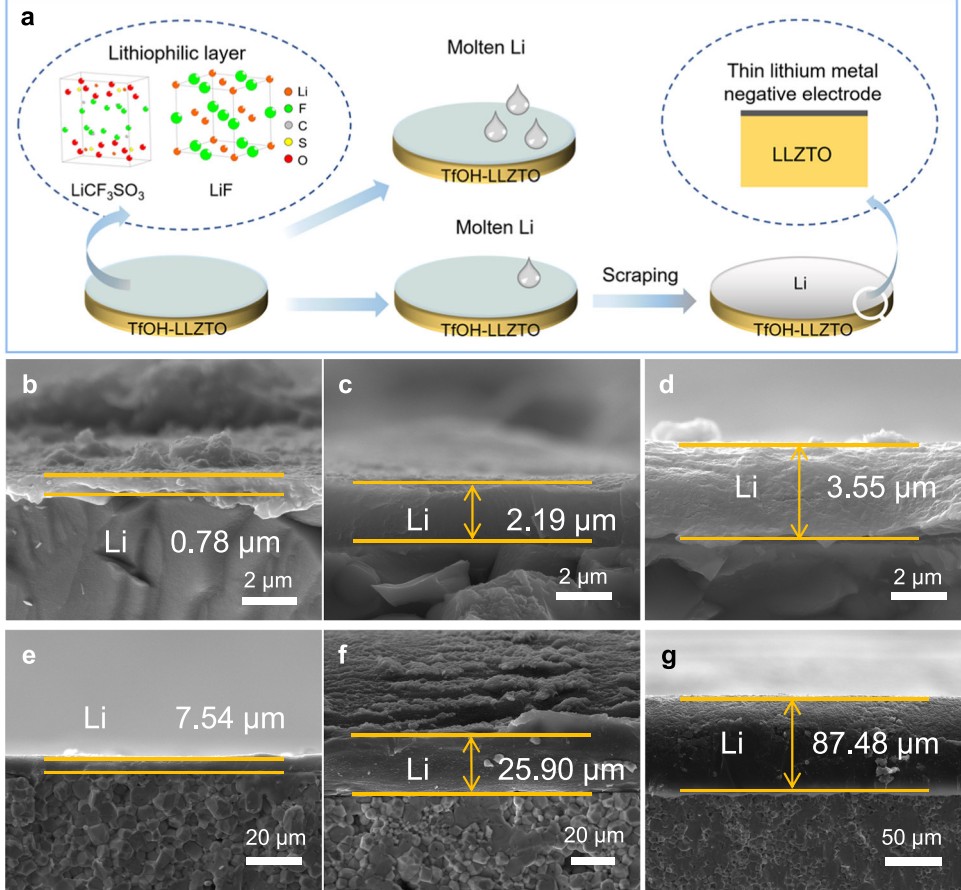

**Fig. 4 | Preparation and characterization of thin Li metal negative electrodes.**
**a** A diagram shows the strategy of accurately controlling the thickness of a lithium metal negative electrode through in situ conversion of a lithiophilic interlayer at the interface and quantitative lithium melt-scraping. **b**–**g** Cross-sectional SEM images of interfaces composed of SSE and lithium metals of 0.78, 2.19, 3.55, 7.54, 25.90, and 87.48 μm.

(Supplementary Fig. S12c, d). It demonstrates the enhanced stability and generalizability of the thickness controllable preparation strategy for thin lithium negative electrodes. Significantly, a thin thickness down to only 0.78 μm could be achieved (Supplementary Fig. S13). Besides, the as-prepared thin lithium layer is uniformly attached across the surface of TfOH-LLZTO, establishing close contact with TfOH-LLZTO.

**Improved stability of QSSLMBs with thin Li metal negative electrodes**

Afterward, quasi-solid-state lithium-metal battery (QSSLMB) was assembled using Li-coated TfOH-LLZTO as the negative electrode/electrolyte, and commercially available LiFePO$_4$ (LFP)- or LiNi$_{0.83}$Co$_{0.11}$Mn$_{0.06}$O$_2$ (NCM)-coated Al foil was directly used as the positive electrode (Fig. 5a). A small amount of lithium hexafluorophosphate liquid electrolyte was applied only at the positive electrode side as a wettability additive for facilitating the infiltration of Li ions inside the positive electrode. Firstly, a Li|TfOH-LLZTO|LFP QSSLMB with an excessive amount of Li (87.48 μm in thickness, N/P ratio of 28.6) was constructed to confirm the improved interface properties of TfOH-LLZTO under full battery operation conditions. The Li|TfOH-LLZTO|LFP cell exhibits specific discharge capacities of 145.2, 138.5, 132.6, 126.3, 119.4, 114.3, and 109.5 mAh g$^{-1}$ at current densities of 0.11 (0.2), 0.26 (0.5), 0.53 (1), 0.79 (1.5), 1.05 (2), 1.32 (2.5) and 1.58 (3) mA cm$^{-2}$ (C), respectively, at 25 °C (Supplementary Fig. S14a). Even after cycling at a high rate of 1.58 mA cm$^{-2}$, the battery still maintains a specific discharge capacity of 144.3 mAh g$^{-1}$ when the current density was reverted to 0.11 mA cm$^{-2}$, clearly showing the high

reversibility of the battery (Supplementary Fig. S14b). Remarkably, this TfOH-LLZTO-based QSSLMB could steadily run for 500 cycles at 1.05 mA cm$^{-2}$ and retain 81% of its initial capacity (122.5 mAh g$^{-1}$) after the test at 25 °C (Supplementary Fig. S15). In addition, an average coulomb efficiency of 99.5% was achieved for this QSSLMB, further confirming its good interfacial stability.

Furthermore, due to the enhanced lithiophilicity of this TfOH-LLZTO electrolyte, precise regulation of the thickness of lithium metal electrodes has been achieved, which enables a high Li metal utilization, i.e., a high energy density of the QSSLMBs, as well as the in-depth investigation of the influence of Li metal thickness on their electrochemical performance.

Thus, the TfOH-LLZTO-based QSSLMBs were assembled with different N/P ratios from 0.1 to 3.4. For this study, a commercial NCM positive electrode with an areal capacity of 1.5 mAh cm$^{-2}$ was adopted to verify the practical viability of these QSSLMBs (i.e., Li|TfOH-LLZTO|NCM). These batteries with various N/P ratios were tested under 0.14 mA cm$^{-2}$ at 25 °C. Although their initial charge-discharge curves show nearly the same discharge specific capacity of ~192 mAh g$^{-1}$ (Supplementary Fig. S16), the batteries with different N/P ratios show distinctly different cycling performances. As shown in Fig. 5b, the cycling life of the QSSLMB with the extremely low N/P ratio of 0.1 (i.e., Li-0.78|TfOH-LLZTO|NCM with 0.78 μm Li metal negative electrode) is fairly short, and the discharge capacity experiences a rapid decrease due to the limited lithium supply. As the N/P ratio increases, the cycling performance of the batteries significantly improves. It is worth noting that the batteries with N/P ratios of 1.0 (i.e., 7.54 μm Li metal negative electrode) and 3.4 (i.e., 25.9 μm Li metal negative electrode) show the

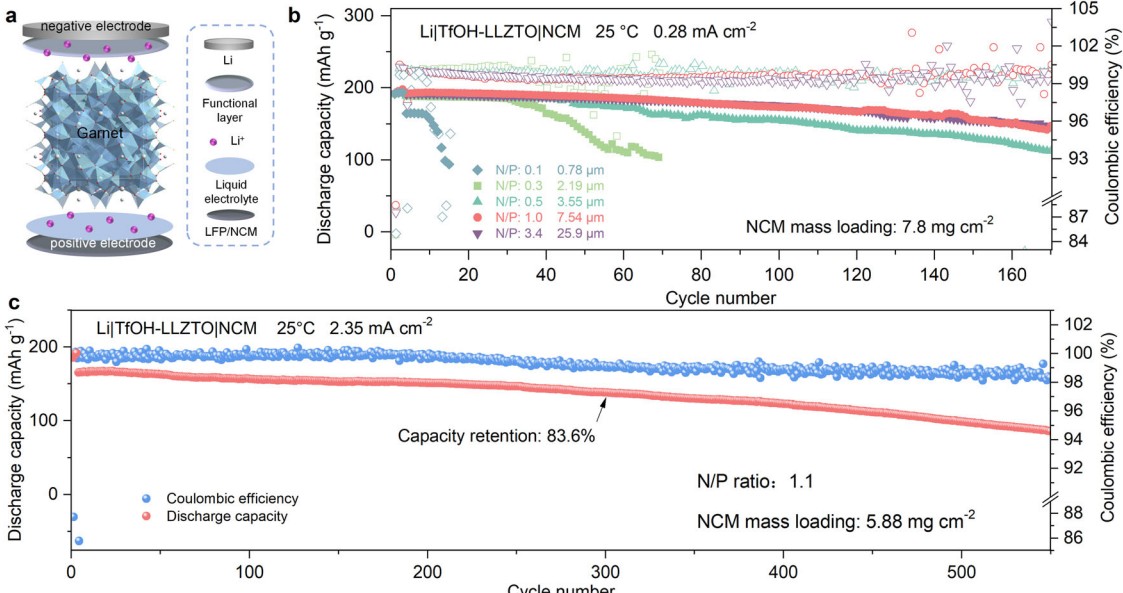

**Fig. 5 | Electrochemical performance of QSSLMBs with thin Li metal negative electrodes. a** Schematic of a solid-state Li cell with TfOH-LLZTO and positive electrode. **b** Cycling performances of Li|TfOH-LLZTO|NCM cells using limited Li metal at various N/P ratios. **c** Cycling performances of Li|TfOH-LLZTO|NCM cells using limited Li metal (7.54 μm) at a N/P ratio of 1.1. Source data for Fig. 5b, c are provided as a Source Data file.

same enhanced cycling performance, with capacity retention up to 76.6% and 76.7% after 170 cycles, respectively, signifying that the thickness of 7.54 μm is the optimal for QSSLMBs with limited lithium. To further verify the cycling stability of the thin lithium metal negative electrode with the thickness of 7.54 μm in QSSLMBs, we examined the long-term cycling performance of the batteries at a high current density at 25 °C (Fig. 5c). Notably, the cells were successfully operated for over 500 cycles at a current density of 2.35 mA cm⁻², with a discharge specific capacity of 138.3 mAh g⁻¹ and a cycle retention rate of 83.6% after 300 cycles. The cyclability of the battery can also be attributed to the chemical stability of the TfOH-modified layer and Li metal.

Reducing the consumption of lithium metal is a significant challenge for the practical application of SSLMBs. The research progress of SSLMBs with garnet-type SSEs is summarized in Supplementary Table S2. Benefiting from the gratifying cycling stability ascribed to the good durability of lithiated Li|TfOH-LLZTO interface, our QSSLMB with a high mass loading of NCM positive electrode and limited Li metal negative electrode with 7.54 μm in this work can stably cycle for more than 500 cycles at 25 °C, representing the leading performance up to now for SSLMBs to our knowledge (Supplementary Fig. S17a)[18,37,46,47] and thus validating the effectiveness of this thin lithium negative electrode strategy enhancing the performance of SSLMBs. Besides, to our knowledge, this remarkably high rate performance and cycling stability also make this TfOH-LLZTO electrolyte much more advantageous than other garnet-type SSEs with various surface-modification layers (Supplementary Fig. 17b)[43,48–63]. In addition, to further verify the applicational feasibility of this technology for SSLMBs, we assembled SSLMBs with solid polymer electrolytes (SPEs) composed of succinonitrile (SN) and lithium bis(trifluoromethanesulfonyl)imide (LiTFSI) between TfOH-LLZTO and positive electrode to improve the interfacial contact (Supplementary Fig. S18). As shown in Supplementary Fig. S19, the assembled battery can be stably cycled for 28 cycles at 0.25 mA cm⁻² (0.5 C), with a capacity retention of 84%, which is much higher than that of SSLMBs using the SPEs on both sides (39%). This again demonstrated the advantage of TfOH-LLZTO in improving battery performance by enhancing contact at the Li|SSE interface.

## Evolution of internal components of thin lithium metals

To further understand the influence of lithium negative electrode thickness on the cycling behaviors of batteries, XPS with Ar sputtering depth profiles was conducted to analyze the detailed composition distribution from the Li metal surface to the Li|TfOH-LLZTO interface of the Li metal negative electrodes with thicknesses of 0.78 μm (Li-0.78|TfOH-LLZTO) and 7.54 μm (Li-7.54|TfOH-LLZTO). Supplementary Fig. S20 shows the detailed composition distribution of Li-0.78|TfOH-LLZTO from the Li surface to the Li|TfOH-LLZTO interface. The F 1s peak at 684.75 eV can be assigned to LiF (Supplementary Fig. S20a), while the new O 1s peak at 529.88 eV corresponds to Li₂O (Supplementary Fig. S20b). In addition, the S 2p peaks at 166.84 eV and 168.09 eV correspond to the Li₂SO₃, and the S 2p peaks at 160.22 eV and 161.37 eV correspond to Li₂S (Supplementary Fig. S20c). The peak intensity of La 3d significantly increased after 1 min of Ar sputtering, and it kept constant after 2 min of Ar sputtering (Supplementary Fig. S20d), indicating that the 2 min Ar sputtering reached the Li-0.78|TfOH-LLZTO interface. These results reveal that LiCF₃SO₃ was converted to LiF, Li₂SO₃, Li₂S, and Li₂O upon contact with Li metal, which consistent with the calculation results (Supplementary Note S2and Supplementary Equation S3). Furthermore, as the Ar sputtering time increases, the peak intensity corresponding to LiF, Li₂O, Li₂S, and LiCF₃SO₃ remarkably increases, while the intensity of Li₂SO₃ slightly decreases (Supplementary Fig. S21). This result indicates that more LiF, Li₂O, Li₂S, and LiCF₃SO₃ species are enriched at the interface of Li-0.78|TfOH-LLZTO, while Li₂SO₃ is dispersed throughout the thin lithium metal. The LiF[29,37,64] and Li₂O[48,65] species, which are stably enriched at the Li|TfOH-LLZTO interface and exhibit high interfacial energy and low electronic conductivity, could effectively inhibit Li dendrite growth.

After the cycling of Li-0.78|TfOH-LLZTO|NCM battery, the interface of Li|TfOH-LLZTO remains LiF-enriched (Supplementary Figs. S22, S23). However, it is worth noting that Li₂O, the primary component before cycling, undergoes complete depletion after cycling, giving rise to the emergence of a new substance, Li₂O₂. Therefore, it is reasonable to speculate that in the lithium-deficient scenario, the rapid consumption of active lithium metal in the negative electrode leads to the

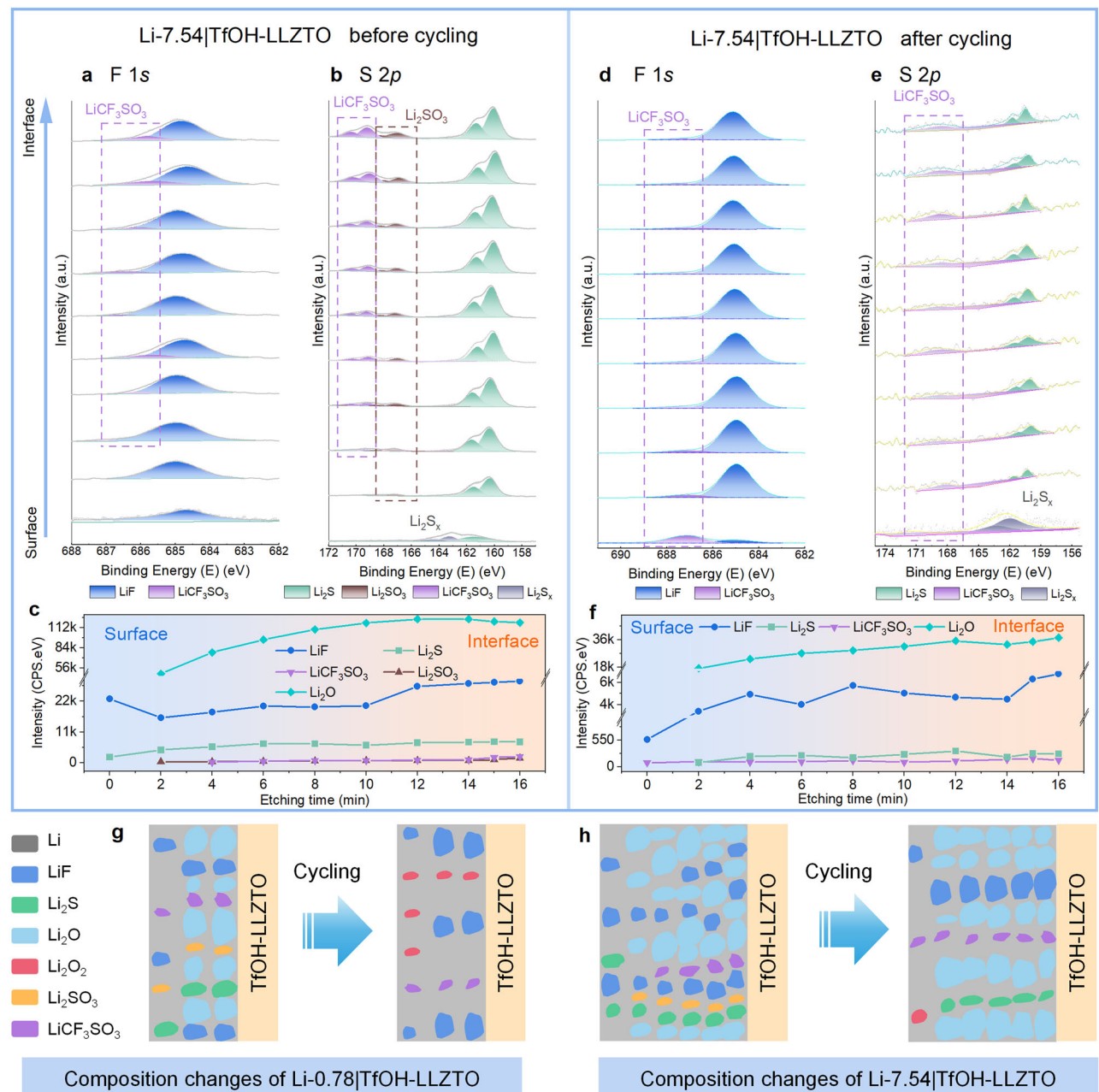

**Fig. 6 | Multi-dimensional characterization of thin Li metal negative electrodes. a, b** XPS spectra of F 1*s* and S 2*p* of Li-7.54 on TfOH-LLZTO of a Li-7.54|TfOH-LLZTO|NCM cell before cycling. **c** Intensity of different components originating from the related peaks in the F 1*s*, S 2*p*, and O 1*s* spectra before cycling. **d, e** XPS spectra of F 1*s* and S 2*p* of Li-7.54 on TfOH-LLZTO of a Li-7.54|TfOH-LLZTO|NCM cell after cycling for 170 cycles under 0.14 mA cm$^{-2}$ at 25 °C. **f** Intensity of different components originating from the related peaks in the F 1*s*, S 2*p*, and O 1*s* spectra after cycling. **g** Schematic of the Li-0.78|TfOH-LLZTO interface composition of a Li-0.78|TfOH-LLZTO|NCM cell before and after cycling. **h** Schematic of the Li-7.54| TfOH-LLZTO interface composition of a Li-7.54|TfOH-LLZTO|NCM cell before and after cycling. Source data for Fig. 6a-f are provided as a Source Data file.

delithiation of $Li_2O$ to supplement lithium ions and maintain battery cycling[66]. $Li_2O_2$, characterized by low ionic and electronic conductivity ($10^{-19}$ S cm$^{-1}$)[67], is a byproduct of $Li_2O$'s delithiation[66]. In addition, the reaction between $Li_2SO_3$ and lithium metal (Supplementary Equation S4) is another possible origin of $Li_2O_2$ (Supplementary Fig. S24). In the lithium-deficient negative electrode, insufficient lithium source can lead to the delithiation and conversion of $Li_2O$, while the poorly reversible conversion of $Li_2O/Li_2O_2$ leads to the sustained consumption of $Li_2O$[68]. Although this conversion can provide lithium ions for cycling in the short term, the deficiency of $Li_2O$, the main components with high surface energy, coupled with the generation of $Li_2O_2$, may

promote the uneven deposition of lithium ions at the interface resulting in the formation of lithium dendrites[48,65]. However, the loose lithium dendrites are easy to break and detach from the interface, losing the transmission channel to become inactive lithium[69]. This, in turn, accelerates the consumption of active lithium, and further deteriorates the kinetics performance of the thin lithium negative electrode.

Then, XPS analysis with Ar sputtering depth profiles was conducted on Li-7.54|TfOH-LLZTO|NCM. In this case, LiF was observed on the surface of Li-7.54 without $LiCF_3SO_3$ (Fig. 6a), which is consistent with the results of S 2*p* (Fig. 6b). In addition, Fig. 6c shows that before

cycling, the distribution of components in Li-7.54 is uneven, with LiF enriched on the surface. However, $Li_2S$, $Li_2O$ (Supplementary Fig. S25), $LiCF_3SO_3$, and $Li_2SO_3$ speceis tend to concentrate at the Li-7.54|TfOH-LLZTO interface. After long-term cycling of Li-7.54|TfOH-LLZTO|NCM, significant changes were observed in the internal composition of Li-7.54, with LiF no longer enriching at the surface (Fig. 6d). At the same time, the signal of $Li_2SO_3$ disappeared (Fig. 6e), indicating a sufficient reaction of $Li_2SO_3$ with Li and the production of $Li_2S$ (Supplementary Equation S5and Supplementary Fig. S26). In Fig. 6f, it can be visually observed that LiF constantly takes up a high content within Li-7.54, which may be due to further decomposition of $LiCF_3SO_3$. At the same time, substances such as LiF, $Li_2S$, $Li_2O$, and $LiCF_3SO_3$ are relatively evenly distributed inside Li-7.54. This indicates that sufficient lithium metal makes the reaction more thorough, and the resulting Li-containing species, such as LiF, $Li_2S$, and $Li_2O$, were stable during cycling and kept a uniform distribution within the Li negative electrode. These characteristics are beneficial for extending the cycle life of batteries.

In addition, the morphological characteristics of lithium metal negative electrodes post-cycling were investigated. Following cycling, Li-0.78, while tightly adherent to LLZTO (Supplementary Fig. S27a), exhibited irregular surface protrusions (Supplementary Fig. S27b) and increased surface roughness compared to its pre-cycling state (Supplementary Fig. S12). In contrast, post-cycling, Li-7.54 maintained a relatively smooth surface and demonstrated robust bonding with LLZTO (Supplementary Fig. S28). This underscores the advantageous cycling performance of Li-7.54, conducive to the uniform deposition of lithium ions.

On the basis of these characterizations, the failure mechanism of the SSLMBs can be proposed. In the lithium-deficient case (i.e., Li-0.78|TfOH-LLZTO), $Li_2O$, the main component apart from Li, undergoes delithiation to compensate for the insufficient lithium source. However, owing to poor reaction reversibility, $Li_2O$ with high surface energy is rapidly and completely consumed, and $Li_2O_2$ with low ionic conductivity is generated, resulting in the uneven deposition of lithium ions and easily detached lithium dendrites, which aggravates the consumption of active lithium. In addition, $Li_2O_2$, produced by $Li_2O$ delithiation and $Li_2SO_3$ decomposition, can significantly compromise battery's cycling performance due to its low ion conductivity (Fig. 6g). In contrast, for the lithium-rich case for Li-7.54|TfOH-LLZTO, complete interfacial reactions lead to the generation of LiF and $Li_2S$, which are characterized by good electronic insulation and high ionic conductivity, respectively, significantly improving the cycling stability of the battery. Besides, $Li_2O$ will not be converted into $Li_2O_2$ in this lithium-rich condition. And such $Li_2O$ species is conducive to the uniform deposition of lithium, and the internal component distribution is more uniform as well, thus further facilitating ionic transport (Fig. 6h).

In addition, the microstructure characteristics of NCM-positive electrode materials in SSLMBs were investigated using focused ion beam-scanning electron microscopy (FIB-SEM) images and EDS. As shown in Supplementary Fig. S29, the positive electrode materials exhibit tight and uniform contact between the components of the positive electrode materials, and there are no cracks inside the positive electrode particles before or after cycling. These phenomena indicate that positive electrode materials have enhanced ion transport performance in SSLMBs. Next, XPS measurements revealed the chemical bonding environment of the cathode-electrolyte interphase (CEI) formed on the NCM positive electrode before and after cycling (Supplementary Fig. S30). In the O 1s spectrum of NCM particles after cycling, peaks belonging to C = O (533.12 eV), C-O (534.34 eV), and P-O (531.26 eV) can be observed (Supplementary Fig. S30a). Peaks of $Li_xPO_yF_z$ (687.67 eV), C-F (689.10 eV), and LiF (686.01 eV) were observed in the F 1s spectrum (Supplementary Fig. S30b). In the P 2p spectrum, two phosphorus peaks, $Li_xPO_yF_z$ (133.33 eV) and $Li_xPF_y$ (134.86 eV), originating from unexpected ionic conductors, were fitted

(Supplementary Fig. S30c)[70]. It indicates that the added lithium hexa-fluorophosphate liquid electrolyte forms a CEI containing F element on the surface of NCM, and at the same time, it undergoes certain decomposition after cycling to generate unexpected $Li_xPO_yF_x$ and $Li_xPF_y$. This decomposition is not conducive to the cycling of the batteries, which may be the reason for the decay of the NCM-positive electrodes.

## Discussion

In summary, we propose an in situ conversion strategy of $Li_2CO_3$ components on the surface of Ta-doped $Li_7La_3Zr_2O_{12}$ (LLZTO) with trifluoromethanesulfonic acid (TfOH) to form a super lithiophilic and electron-blocking TfOH-modified layer over LLZTO, which enables the precise thickness control of Li metal negative electrodes in the range of 0.78 μm to 30 μm. A symmetric cell based on this TfOH-LLZTO SSE achieved stable cycling for 800 h at a high current density of 1.0 mA $cm^{-2}$ at 25 °C. When it is coated with a Li negative electrode with a thickness of 7.54 μm and coupled with a commercial NCM positive electrode to assemble a QSSLMB full cell, a long cycling life over 500 cycles is achieved at a high current density of 2.35 mA $cm^{-2}$ at a low N/P ratio of 1.1. Moreover, through multi-scale characterizations of thin lithium negative electrodes, we, clarify the multi-dimensional compositional evolution and failure mechanisms in the lithium-deficient and -rich regions (0.78 μm and 7.54 μm) of lithium negative electrodes, on its surface, inside it, or at the Li|LLZTO interface. It was found that the insufficient interfacial reaction in the lithium-deficient case is the key factor for the occurrence of severe side reactions and even battery failure. This strategy of constructing a thin lithium metal layer on SSEs provides a practical solution and key insights for the application of the next generation of high-performance Li metal batteries.

## Methods

### Materials synthesis

The synthesis of LLZTO garnet electrolyte used a conventional solid-state reaction method[71,72]. Stoichiometric amounts of $Li_2CO_3$ (99.99%, Aladdin), $La_2O_3$ (99.99%, Macklin), $ZrO_2$(99.99%, Aladdin), and $Ta_2O_5$ (99.99%, Macklin) were mixed in isopropanol (99%, Macklin). In addition, ≈10 wt. % excess $Li_2CO_3$ was added to compensate for the volatilization of Li at high-temperature calcination. The mixed starting materials were ball milled at 500 rpm for 24 h (ball mill model QM-QX2, Nanjing Nanda Instrument), and then calcined at 950 °C for 12 h. The ball mill jar is made of zirconia and has a volume of 100 mL. The ball mill beads are made of zirconia. The mass ratio of ball milled beads to raw materials is 1.5:1. The ball milling interval is 5 min. The obtained powders were sieved through 200 grits and then pressed into pellets. The electrolyte powder was pressed and formed under a pressure of 10 MPa for 5 min using a circular mold made of Cr12MoV (with an internal diameter of 16 mm). After the pressing program is completed, an electrolyte sheet with a diameter of 16 mm and a thickness of approximately 1 mm is obtained. The pressing process is carried out at 25 °C in a glove box filled with nitrogen gas. Then, the green bodies were sintered at 1150 °C for 6 h in the air. The heating rate and cooling rate are both 5 °C per minute below 900 °C and 3 °C per minute above 900 °C. To avoid the Li loss in the sintering process, all the green bodies were covered with mother powders. After sintering, the samples were first polished with 400, 800, and 2000 mesh sandpaper successively. The thickness of the pellets is ≈600 microns. These pellets are stored in a glove box filled with argon for future use.

### Assembly of solid-state cells

In a typical Li symmetric cell assembly, Li foil (99.99%, Adamas-beta) was first melted using a hotplate (JF-956S, JFTOOLS) at ≈300 °C and then coated on the LLZTO surface with Doctor blade. The QSSLMBs were assembled in CR2032-type coin cells by contacting a lithium metal negative electrode, the as-prepared LLZTO SSE, and a LiFePO4

(Homemade) or $LiNi_{0.83}Co_{0.11}Mn_{0.06}O_2$ (NCM, RONBAY TECHNOLOGY) positive electrode. The raw materials for $LiFePO_4$ are $LiOH$ (99.99%, Aladdin), $FeSO_4$ (98%, Macklin), and $H_3PO_4$ (AR, Macklin). These raw materials were subjected to hydrothermal reaction (120 °C for 6 h) in a high-pressure reaction vessel (LC-KH-25, LICHEN) according to the mole ratio ($LiOH:FeSO_4:H_3PO_4$ = 3:1:1). Then, 2 g of carbon black (AR, Aladdin) was added to 100 g of the obtained material and ball milled in a 100 mL agate ball milling jar (250 rpm for 2 h, ball mill model QM-QX2, Nanjing Nanda Instrument). The ball-milling beads were agate beads. The mass ratio of ball-milling beads to raw materials is 1.5:1. The ball-milling interval is 5 min. The above product was then sintered in a high-temperature tube furnace (argon environment, 500 °C for 3 h, OTF-1200X-S, Kejing) to obtain $LiFePO_4$. The heating rate and cooling rate are both 3 °C per minute. The positive electrode was composed of 80 wt% $LiFePO_4$ or NCM, 10 wt% carbon black (AR, Macklin), and 10 wt% polyvinylidene difluoride (99.92%, Kelude) binder. The positive electrode composite was added into anhydrous acetonitrile (99.9%, Macklin) and then coated on an aluminum substrate with Doctor blade (single sided coating). Subsequently, the acetonitrile solvent was removed at 60 °C under vacuum, and the electrode was punched with a diameter of 12 mm using manual cutting and slicing machine (MSK-T10, Kejing). During the battery preparation process, 3 μL of lithium hexafluorophosphate electrolyte (LB-037, 1 M $LiPF_6$ in diethyl carbonate:ethylene carbonate:methyl ethyl carbonate = 1:1:1 Vol%, purity 99.9%, moisture content less than 50 ppm, DoDoChem) was added between the positive electrode and the solid electrolyte. All the cells were assembled in CR2032-type coin (the case and spring material are made of 304 stainless steel) cells in an argon-filled glove-box.

## Materials characterization

Collect samples in a glove box filled with argon gas and seal and transport them in a vacuum sealed plastic bag at 25 °C. The morphology of as-prepared samples was characterized by scanning electron microscopy (SEM) (JEOL, JSM-7600F). X-ray diffraction (XRD, Bruker D8 Advanced, Germany) was applied to evaluate crystalline structure. Raman spectra were obtained on Jobin-Yvon LabRAM HR-800. FTIR spectra were obtained on Bruker Vector-22 FTIR spectrometer in the range of 4000-400 $cm^{-1}$. The X-ray photoelectron spectroscopy(XPS) test was performed on an ESCALAB 250 spectrometer. Image J is used to measure the thickness of the lithium layer in the image[73].

## Electrochemical measurements

The CCD value of Li‖Li symmetric cell was determined under an initial current density of 0.1 mA $cm^{-2}$ with an increasing step of 0.1 mA $cm^{-2}$ with a capacity of 0.2 mAh $cm^{-2}$. For cell performance with LFP positive electrode, the rate capability of SSLMBs was measured at 0.2, 0.5, 1, 1.5, 2, 2.5, and 3 C (1 C = 170 mA $g^{-1}$). All the batteries with LFP were tested between 2.5 and 4.0 V at 25 °C. For the NCM positive electrode (1 C = 200 mA $g^{-1}$), the batteries were cycled at 0.2 C between 2.7 and 4.3 V at 25 °C. Electrochemical impedance spectroscopy (EIS) spectra of the cell were obtained from 10 MHz to 6 MHz with a polarization potential of 10 mV and 10 data points per frequency decade. A 30 s open-circuit voltage measurement was carried out prior to measuring to secure that the system is in a relaxed state. Applied signal is galvanostatic. All the electrochemical tests were carried out using a BioLogic VSP-300 electrochemical workstation and Land multichannel battery test system. The electrochemical performance test is conducted in an environmental chamber with a temperature maintained at 25 °C.

## DRT analyses

DRT is a model-free analysis of impedance evolution[74]. DRT spectrum was transformed from Nyquist plots of EIS results based on the frequency domain[30,75]. The DRT results in this article are generated by DRT tools[31], which run on MATLAB 2021b to satisfy the DRT practices for EIS deconvolution. The Gaussian process was used to realize DRT deconvolution. We selected both real and imaginary parts together "Combined Re-Im Data" of the EIS data to be used for the computation of the DRT. Then, we selected to fit without inductance for treating the inductive features. We chose 1st order as the derivative used in the penalty, which is the norm of the first-order derivative of γ(lnτ). The regularization parameter λ and sample number were selected as 1E-3 and 10000 respectively.

## Tafel test

The Tafel equation is expressed as follows[37]: $\eta = RT\ln i_O / (\alpha F) - RT\ln i / (\alpha F)$ ($\eta$, voltage polarization; $R$, gas constant; $F$, Faraday's constant; $T$, system temperature (333.15 K); $\alpha$, charge transfer coefficient; $i$, exchange current density). In the strong polarization region, a linear relationship is observed between the natural logarithm of the current (ln $i$) and the polarization ($\eta$). The Allen-Hickling plots are constructed based on the Butler-Volmer equation presented below[76]: $\ln[I / (1 - e^{F\eta/(RT)})] = \ln i_O - \alpha F\eta / (RT)$. Furthermore, the corresponding charge transfer resistance ($R_{CT}$) can be determined using the exchange current density ($i_O$), and the relationship outlined below[37]: $R_{CT} = RT / (Fi_O)$.

## Density functional theory (DFT) calculations

Density functional theory calculations[77,78] were performed using the Vienna ab initio simulation package (VASP), employing the plane-wave basis sets and the projector augmented-wave method[79,80]. The exchange-correlation potential was approximated using a generalized gradient approximation (GGA) with the Perdew-Burke-Ernzerhof (PBE) parametrization[81]. The van der Waals correction was implemented using Grimme's DFT-D3 model[82]. An energy cutoff of 450 eV was applied, and structural relaxation was performed until the maximum force on each atom dropped below 0.03 eV/Å. The energy convergence criterion was set to $10^{-5}$ eV.

The interfacial models of $Li|Li_7La_3Zr_2O_{12}$, $Li|LiCF_3SO_3$, $Li|LiF$ and $Li|Li_2CO_3$, were constructed by the low-energy surface of Li(001), $Li_7La_3Zr_2O_{12}$(001), $LiCF_3SO_3$(001), LiF(001) and $Li_2CO_3$(001) slabs. $4 \times 4 \times 1/1 \times 1 \times 1$, $3 \times 3 \times 1/1 \times 1 \times 1$, $4 \times 4 \times 1/3 \times 3 \times 1$, and $3 \times 3 \times 1/2 \times 2 \times 1$ supercells were used to construct Li(001)|$Li_7La_3Zr_2O_{12}$(001), Li(001)|$LiCF_3SO_3$(001), Li(001)|LiF(001) and Li (001)|$Li_2CO_3$(001) interface layers, respectively. Structural parameters obtained after the optimization are. $W_{ad}$ can be acquired according to the following equation: $W_{ad} = (E_{Li} + E_{X\text{-slab}} - E_{interface}) / S$, where $E_{interface}$, $E_{Li}$, and $E_{X\text{-slab}}$ (X = $Li_7La_3Zr_2O_{12}$(001), $LiCF_3SO_3$(001), LiF (001) and $Li_2CO_3$ (001)) are referenced to the total energy of interfacial supercell, isolated Li (001) and X surface slab, respectively, and $S$ refers to interfacial area[83].

The contact angle of these interfaces was calculated by Young-Dupré equation[84] as following: $W_{ad} = \sigma_{Li}$ $(1 + cos\theta)$, ($W_{ad}$, work of adhesion for the interface; $\sigma_{Li}$, surface energy of metal lithium; $\theta$, contact angle).

## Data availability

The authors declare that the data supporting the findings of this study are available within the paper and its supplementary information files. Source data are provided in this paper.

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

## Acknowledgements

The authors thank Hui-Ming Cheng (Chinese Academy of Science) for guidance and suggestions in this article. The authors acknowledge Professor Rongchao Jin (Carnegie Mellon University) for guidance in this article. The authors acknowledge Xinyuan He (Central South University) for her help in preparing materials. This work was financially supported by the National Natural Science Foundation of China (51902347 to J.Z., 52202338 to X.W., 22179093 and 22379111 to J.L.) and Postgraduate Scientific Research Innovation Project of Hunan Province (CX20230183 to W.J.).

## Author contributions

Ji, W., Investigation, Data curation, Visualization, Writing-original draft. Luo, B., Conceptualization; Wang, Q., Methodology; Yu, G., Methodology. Zhang, Z., Investigation. Tian, Y., Investigation; Zhao, Z., Investigation; Zhao, R., Investigation. Wang, S., Data curation. Wang, X., Investigation, Data curation, Writing-review & editing, Funding acquisition. Zhang, B., Validation. Zhang, J., Conceptualization, Writing-review & editing, Funding acquisition. Sang, Z., Visualization, Investigation. Liang, J., Investigation, Conceptualization, Writing-review & editing, Funding acquisition.

## Competing interests

The authors declare no competing interests.
