## [Peer Review File · Nature Communications]

REVIEWER COMMENTS

Reviewer #1 (Remarks to the Author):

In this manuscript, the authors adopted $\text{CF}_3\text{SO}_3\text{H}(\text{TfOH})$ solution to in-situ transform the Li_2CO_3 layer on the surface of LLZTO into LiCF_3SO_3 to improve the interface contact of Li/LLZTO, and the thickness of the metal lithium layer was controlled by scraper. Finally, they claimed an “ASSLMB full cell” was assembled by coupling with a commercial NCM cathode. However, there are a considerable amount of reports in recent years on the physical or chemical surface treatment of LLZO surface to promote its infiltration with molten lithium, even the direct wetting of LLZO with molten Li. In addition, the way of controlling the thickness of the metal lithium layer by scraper is hard to be transplanted and scaled up, especially when the infiltration is determined by the amount of Li_2CO_3 on the surface of LLZTO. And the assembled battery shouldn't be called “all-solid-state battery” due to the using of liquid electrolyte on the positive side. As a result, the original innovation of this paper might not be enough to match the requirements of this journal.

Reviewer #2 (Remarks to the Author):

In my opinion it is a well-written and informative paper, in which the Authors proved that by the developed facile interface engineering it is possible to significantly enhance the Li/Li₇La₃Zr₂O₁₂ interface, making it suitable for the all solid-state lithium-metal batteries (ASSLMBs). There is suitable novelty present, as well as the conclusions are of practical importance. The proposed treatment of the garnet surface with trifluoromethanesulfonic acid resolves the most important issues associated with presence of the surface Li_2CO_3 , creating at the same time beneficial LiCF_3SO_3 and LiF phases, forming a lithiophilic layer. This, in turn, enables preparing well-attached Li layers, even as thin as 0.78 μm . Importantly, the Authors proved validity of their approach in the manufactured ASSLMBs having different N/P ratio. The degradation mechanism of those cells was studied, and explained in details. There are numerous experimental data provided and appropriately commented. I think that the conclusions are fully-supported, and are of interests for the readers. Consequently, I believe that the submission can be accepted for publication, with only minor comments, as written below.

1) Regarding the TfOH-LLZTO interface (Fig. 1), top-view SEM/EDS data would be of interest, as homogeneous distribution of F and S elements could be better seen.

2) The Authors are asked to provide more precise evaluation of thickness of the Li layers (Fig. 4), as e.g. no error values are given. Also, there are some features/unevenness visible from the top-side view in Fig. 4, which for the thin layers may be somewhat problematic. This is not commented.

3) It is not clear how the EIS/DRT data were treated to prepare Figs. S6b and d. For example, how many spectra were recorded?

4) The same energy units could be provided in Fig. S17 and Note S2.

5) Ionic liquid was(?) used in the full cell assembly procedure (Fig. 5a). This should be explained in details. Microstructural characterization of the cathode side of the full cells could be also shown.

6) Microstructural data of the anode/electrolyte interface after electrochemical tests should be added, if possible.

Response to Reviewers' Comments

Response to Reviewer 1 (Pages 1-14)

Response to Reviewer 2 (Pages 11-27)

Response to Reviewer 1

Reviewer #1 (Revision marked with red in the revised MS)

Comment: *In this manuscript, the authors adopted $CF_3SO_3H(TfOH)$ solution to in-situ transform the Li_2CO_3 layer on the surface of LLZTO into $LiCF_3SO_3$ to improve the interface contact of Li/LLZTO, and the thickness of the metal lithium layer was controlled by scraper. Finally, they claimed an "ASSLMB full cell" was assembled by coupling with a commercial NCM cathode. However, there are a considerable amount of reports in recent years on the physical or chemical surface treatment of LLZO surface to promote its infiltration with molten lithium, even the direct wetting of LLZO with molten Li. In addition, the way of controlling the thickness of the metal lithium layer by scraper is hard to be transplanted and scaled up, especially when the infiltration is determined by the amount of Li_2CO_3 on the surface of LLZTO. And the assembled battery shouldn't be called "all-solid-state battery" due to the using of liquid electrolyte on the positive side. As a result, the original innovation of this paper might not be enough to match the requirements of this journal.*

Comment 1. *However, there are a considerable amount of reports in recent years on the physical or chemical surface treatment of LLZO surface to promote its infiltration with molten lithium, even the direct wetting of LLZO with molten Li.*

Response: Firstly, we would like to express our sincere gratitude to the reviewer for the insightful comments and suggestions, which are very helpful for us to improve the quality of our work. Below, we provide our arguments on the novelty of our work compared to the literature reports.

Indeed, extensive previous research has been conducted on modifying the surface of $Li_7La_3Zr_2O_{12}$ (LLZO) to enhance its wettability towards lithium; and molten lithium can directly wet these reported modified LLZO to some extent. By far, most of the reported interface modification technologies are

associated with surface chemical treatment (*Angew. Chem. Int. Ed. Engl.*, 2020, 59, 12069), physical surface treatment (*Adv Energy Mater.*, 2023, 13, 2300165), artificial interface layer construction (*Nat. Mater.*, 2017, 16, 572., *Sci. Adv.*, 2017, 3, e1601659), and composite lithium metal design (*Energy Environ. Sci.*, 2023, 16, 1049., *Angew. Chem. Int. Ed.*, 2023, 63, e202315856). The common feature of these methods is to enhance the wettability of the interface, thereby improving the interface contact. However, these works only enhance the materials' wettability towards lithium to some extent, and achieving super lithiophilicity is still extremely hard. This leads to significant difficulties in precisely tuning the thickness of lithium anode, especially in constructing the ultra-thin lithium anodes.

Different from the previous works, our current work reports a super lithiophilic layer through controlled air exposure and surface treatment for the first time. With experimental investigation assisted by density functional theory simulations, we successfully achieved a wetting angle down to approximately 0°, demonstrating a super-wettability, which is much superior to previous works (*e.g.*, *Nat. Mater.*, 2017, 16, 572., *Sci. Adv.*, 2017, 3, e1601659., *Energy Environ. Sci.*, 2023, 16, 1049., *Angew. Chem. Int. Ed. Engl.*, 2020, 59, 12069., *Angew. Chem. Int. Ed. Engl.*, 2021, 60, 3781., and *Angew. Chem. Int. Ed.*, 2023, 63, e202315856).

With our new methodology, three major goals have been achieved, which makes this work stand out from the previous ones. First, this super lithiophilic interface allowed us to precisely control the thickness of lithium metal anodes through simple physical scraping, ranging from 0.78 to 30 μm, which has rarely been achieved before.

Second, apart from the ultra-thin lithium anode itself, this technology also enabled the in-depth and multi-scale characterization of the material, which is essential for thoroughly elucidating the multi-dimensional compositional evolution and failure mechanisms of lithium-deficient and lithium-rich regions (0.78 μm and 7.54 μm) on its surface, within and at the Li/LLZTO interface. This has rarely been achieved in previous studies, and it provides valuable insights into the battery operation behaviors.

Third, as for the battery performance, our lithium metal anode with the optimal thickness of 7.54 μm only requires a very low negative-to-positive electrode capacity ratio of 1.1, when integrated with a commercial LiNi_{0.83}Co_{0.11}Mn_{0.06}O₂ cathode to assemble a full cell. It also demonstrated an ultra-long

cycling life of over 500 cycles at a high current density of 2.35 mA cm⁻². Such superior performance is directly associated with the high tunability in lithium anode thickness, again proving the effectiveness of this technology in enhancing the performances of solid-state batteries (SSBs).

Overall, we hope the above points show clear differences from the previous one in terms of materials design and functioning mechanisms, which leads to much superior battery performance and valuable insights into the compositional/structural evolution during the battery operation.

Comment 2. *In addition, the way of controlling the thickness of the metal lithium layer by scraper is hard to be transplanted and scaled up, especially when the infiltration is determined by the amount of Li₂CO₃ on the surface of LLZTO.*

Response: We thank the reviewer for providing this valuable comment, which is helpful in improving the quality of our work. For LLZTO-based oxide solid-state electrolytes, lithium needs to be melted to form good contact, which limits the use of transferable lithium anodes such as ultra-thin lithium foils (e.g. *Sci. Adv.*, 2022, 8, eabq0153., *Nat. Commun.*, 2022, 13, 1883, and *Nat. Commun.*, 2021, 12, 176). Constructing a super-lithiophilic layer combined with physical scraping coating offers a relatively simple, controllable, and economical approach to fabricating ultra-thin lithium in this context. Therefore, we selected the widely adopted and technically mature scraping method, as demonstrated in previous works (e.g. *Nat. Energy*, 2018, 3, 560, *Sci. Adv.*, 2020, 6, 8641, and *Nat. Commun.*, 2020, 11, 582-591), to prepare an ultra-thin lithium layer on a thin piece of LLZTO solid-state electrolyte (SSE). This composite anode/electrolyte is then directly coupled with a commercial cathode to assemble a full battery. This technology might have the potential to be transplanted to other SSE surfaces with similar characteristics. However, as this work is a “proof-of-concept” demonstration of achieving precise thickness control of lithium metal anodes by constructing an ultra lithiophilic layer over LLZTO, we only used this type of SSE as the objective for our investigation here. The reviewer's comments align with our next research objective: to develop a low-temperature, non-melting method for constructing a superior contact interface between lithium metal and oxide solid-state electrolytes. Achieving this would enable the industrial application of ultra-thin lithium foils.

As for the challenge of scaling up this technology, we have conducted additional experiments on larger

SSEs (1.43 cm in diameter compared to 1.2 cm in the work) with precise control of the thickness of lithium metal anodes in the revised version. The results indicate that the thickness of the lithium metal anodes can still be accurately measured on these larger SSE surfaces. This indicates that our method has a certain degree of generalizability.

Additionally, we conducted experiments on LLZTO exposed to air for different durations, analyzing the results using SEM, EDS, and Raman spectroscopy. We found that the lithium carbonate layers formed on LLZTO surface show different morphologies depending on the exposure time. After treatment with TfOH acid, these layers were converted into TfOH-modified layers with different morphological characteristics. The experiments demonstrated that a continuous and dense TfOH-modified layer could be achieved after 24 hours of air exposure, followed by TfOH acid treatment. This indicates that by controlling the exposure time and surface treatment, the formation of TfOH-modified layers can be effectively regulated. To more clearly express this point, we have made corresponding revisions to the revised Manuscript and Supplementary Information, as follows:

On Page 5-6 in the revised Manuscript:

“...Additionally, we investigated the composition and microstructure of the Li_2CO_3 layer over LLZTO surface after being exposed to air for 1 h, 12 h, 24 h, along with acid treatment with TfOH solution, respectively. As shown in Figure S2, Li_2CO_3 were observed on the surface of LLZTO after exposure to air, but Li_2CO_3 no longer existed after the TfOH treatment. It indicates that TfOH-treatment can completely convert the surface Li_2CO_3 layer into the desired lithiophilic layer. Moreover, Figure S3 shows the surface microstructure of LLZTO under different exposure times in the air, revealing an increase in Li_2CO_3 coverage with prolonged exposure time. Notably, after 24 h of air exposure, the LLZTO surface was fully covered with Li_2CO_3 , forming a continuous Li_2CO_3 layer that is conducive to the subsequent formation of a continuous TfOH-modified layer following acid treatment. As anticipated, significant differences in surface microstructure were observed among TfOH-treated samples subjected to different exposure durations, as evidenced in Figure S4. Specifically, compared to specimens exposed for 1 h and 12 h, LLZTO surfaces exposed for 24 h displayed the formation of a continuous TfOH-modified layer following acid treatment with TfOH solution.

Therefore...”

On Page 7 in the revised Manuscript:

“...A minor peak located at around 23° and 29°, corresponding to Li_2CO_3 , appears on air-LLZTO after 24 h exposure to the air.”

On Page 16 in the revised Manuscript:

“...The thickness assessment and corresponding standard deviation are depicted in Figure S11. Additionally, to further confirm the consistency of this technology for fabricating ultra-thin lithium layers, a larger piece of LLZTO (1.43 cm in diameter) was adopted and coated with ultra-thin lithium metal layers of 0.78 μm and 7.54 μm thick, as depicted in Figure S12a, b. The morphology characteristics of the lithium metal layer from cross-view SEM images show a similar morphology to that of the smaller size (1.2 cm in diameter). In the top-view SEM images, the surfaces of these ultra-thin lithium layers are smooth and uniform (Figure S12c, d). It demonstrates the superior stability and generalizability of the thickness controllable preparation strategy for ultra-thin lithium anodes.”

On Page 16 in the revised Supplementary Information:

Figure S12. Cross-view SEM images of interfaces composed of LLZTO and lithium metals of (a) 0.78

μm and (b) $7.54 \mu\text{m}$. Cross-view SEM images of interfaces composed of LLZTO and lithium metals of (c) $0.78 \mu\text{m}$ and (d) $7.54 \mu\text{m}$.

On Page 6-8 in the revised Supplementary Information:

Figure S2. (a) Raman spectra of LLZTO exposed to air for 1 h (as shown in the figure above) and followed by TfOH surface treatment (as shown in the figure below). (b) Raman spectra of LLZTO were exposed to air for 12 h (as shown in the figure above) and followed by TfOH surface treatment (as shown in the figure below). (c) Raman spectra of LLZTO exposed to air for 24 h (as shown in the figure above) and followed by TfOH surface treatment (as shown in the figure below).

Figure S3. (a) Top-SEM, (b) EDS of elemental C, O, La, and Zr, and (c) EDS spectra of LLZTO exposed to air for 1 h. (d) Top-SEM, (e) EDS of elemental C, O, La, and Zr, and (f) EDS spectra of LLZTO exposed to air for 12 h. (g) Top-SEM, (h) EDS of elemental C, O, La, and Zr, and (i) EDS spectra of LLZTO exposed to air for 24 h.

Figure S4. (a) Top-SEM, (b) EDS of elemental F, S, La, and Zr, and (c) EDS spectra of LLZTO exposed to air for 1 h followed by TjOH surface treatment. (d) Top-SEM, (e) EDS of elemental F, S, La, and Zr, and (f) EDS spectra of LLZTO exposed to air for 12 h followed by TjOH surface treatment. (g) Top-SEM, (h) EDS of elemental F, S, La, and Zr, and (i) EDS spectra of LLZTO exposed to air for 24 h followed by TjOH surface treatment.

Comment 3. And the assembled battery shouldn't be called "all-solid-state battery" due to the using

of liquid electrolyte on the positive side.

Response: We thank the reviewer for this comment. To address this, we have (1) changed “all-solid-state battery” to “quasi-solid-state battery” for the one with liquid electrolyte in the cathode, and (2) assembled all-solid-state batteries using our ultra-thin lithium/LLZTO composite anode/SSE and polymerized solid-state electrolyte in the cathode, the performance of which was tested and discussed in the revised manuscript. The specific amendments are as follows.

Firstly, as our study focuses on constructing a highly lithiophilic layer at the electrolyte/lithium metal interface, preparing thin lithium anodes, and investigating the interfacial evolution mechanisms, it is crucial for us to ensure that the performance of the ultra-thin lithium coated on the SSE could be truly revealed. The unreliable interfacial contact at the cathode/SSE interface, which is not the main research objective in this study, may become the real bottleneck that limits the overall battery performance and thus leads to “fake” interpretation of the battery performance results. Therefore, we adopted the commonly used approach of adding a small amount of liquid electrolyte (*i.e.*, an ionic liquid) at the commercial cathode/SSE interface to optimize the interface contacts on the cathode side, to better reveal the true performance of the lithium anode side. A similar method has also been adopted in other recent works, *e.g.*, *Nat. Mater.*, **2023**, 22, 1136; *Sci. Adv.*, **2022**, 8, 0153; *Nat. Commun.*, **2024**, 15, 3586; *Nat. Commun.*, **2023**, 14, 782; *Nat. Commun.*, **2022**, 13, 1883; *Nat. Commun.*, **2021**, 12, 176; *Nat. Commun.*, **2020**, 11, 3716; *J. Am. Chem. Soc.*, **2022**, 144, 2179; *Adv. Mater.*, **2024**, 36, 202308493; *Adv. Mater.*, **2023**, 36, 2308275; *Energy. Environ. Sci.*, **2022**, 15, 1325.

Despite this, we indeed agree with the reviewer that calling such a battery “all-solid-state” is not accurate. Considering this, we conducted additional experiments on constructing all-solid-state batteries using polymerized solid-state electrolytes at the cathode side to improve the interfacial contact between the cathode and LLZTO. This battery has been cycled stably for 28 cycles before its failure. This demonstrates that our ultra-thin lithium process can be applied in all-solid-state batteries without any liquid components, and in the meantime, indicates that the cathode/SSE interfacial contact limits the battery performance in this scenario, as aforementioned. Of course, further optimization of the cathode/SSE interface is still necessary for practical applications, and we believe that better solutions will be made available in the future.

On the basis of the above discussions, we have made corresponding revisions to the revised Manuscript and Supplementary Information, as follows:

On Page 2 in the revised Manuscript:

“...When this lithium metal anode is coupled with a commercial $\text{LiNi}_{0.83}\text{Co}_{0.11}\text{Mn}_{0.06}\text{O}_2$ cathode to assemble a quasi-solid-state lithium-metal battery (QSSLMB), ...”

“...solid-state lithium-metal batteries (SSLMBs) by constructing an ultra-thin lithium anode, deepens the understanding of the failure mechanisms, and provides optimizing avenues for SSLMBs.”

On Page 3 in the revised Manuscript:

“Therefore, solid-state lithium-metal batteries (SSLMBs) stand as a state-of-the-art candidate for...”

“However, the practical application of SSLMBs...”

“...but also significantly compromises the energy density of SSLMBs.”

“...investigation of the intrinsic operation/failure mechanism of SSLMBs.”

“...understanding the operation/failure mechanism of SSLMBs.”

On Page 4 in the revised Manuscript:

“...adopted in SSLMBs due to their intrinsically poor contact with typical SSEs.”

“...over SSLMBs is not only an essential factor for enhancing the performance SSLMBs...”

“...lithium metal and improves the energy density and cycling stability of SSLMBs.”

“...practical application of SSLMBs.”

On Page 17 in the revised Manuscript:

“Improved stability of QSSLMBs with ultra-thin Li metal anodes”

“Afterwards, quasi-solid-state lithium-metal battery (QSSLMB)...”

On Page 18 in the revised Manuscript:

“...a Li/TfOH-LLZTO/LFP QSSLMB...”

“...this TfOH-LLZTO-based QSSLMB could...”

“...was achieved for this QSSLMB...”

“...a high energy density of the QSSLMBs...”

“Thus, the TfOH-LLZTO-based QSSLMBs...”

“...verify the practical viability of these QSSLMBs...”

“As shown in Figure 5b, the cycling life of the QSSLMB...”

On Page 19 in the revised Manuscript:

“...is the optimal for QSSLMBs...”

“...thickness of 7.54 μm in QSSLMBs...”

“...for the practical application of SSLMBs.”

“The research progress of SSLMBs...”

“...our QSSLMB with a high mass...”

“...up to now for SSLMBs to...”

“...strategy enhancing the performance of SSLMBs.”

On Page 20 in the revised Manuscript:

“Figure 5. Electrochemical performance of QSSLMBs with ultra-thin Li metal anodes.”

On Page 22 in the revised Manuscript:

“On the basis of these characterizations, the failure mechanism of the SSLMBs...”

On Page 25 in the revised Manuscript:

“...with a commercial NCM cathode to assemble a QSSLMB...”

On Page 26 in the revised Manuscript:

“The QSSLMBs were assembled in...”

“...the rate capability of SSLMBs...”

On Page 19 in the revised Manuscript:

“...In addition, to further verify the applicational feasibility of this technology for SSLMBs, we assembled SSLMBs with solid polymer electrolytes (SPEs) composed of succinonitrile (SN) and lithium bis(trifluoromethanesulfonyl)imide (LiTFSI) between TfoH-LLZTO and cathode to improve the interfacial contact (Figure S17). As shown in Figure S18, the assembled battery can be stably cycled for 28 cycles at 0.5 C, with a capacity retention of 84%, which is much higher than that of SSLMBs using the SPEs on both sides (39%). This again demonstrated the superiority of TfoH-LLZTO in improving battery performance by enhancing contact at the SSE/Li interface.”

On Page 21-22 in the revised Supplementary Information:

Figure S17. Optical photos of the SPE on the surface of LLZTO before and after polymerization.

Figure S18. Cycling performances of NCM/SPE/TfOH-LLZTO/Li and NCM/SPE/Li cells at 50 °C. Voltage profiles of (b) NCM/SPE/TfOH-LLZTO/Li and (c) NCM/SPE/Li cell at 50 °C.

Comment 4. As a result, the original innovation of this paper might not be enough to match the requirements of this journal.

Response: We'd like to point out the major innovation in this work, which lies in presenting a very simple but effective technology in constructing a super lithiophilic layer over LLZTO SSE, which is essential for achieving exceptionally high thickness control of lithium metal anodes. This directly leads to not only an obvious enhancement in battery performance but also an in-depth understanding of multi-dimensional compositional evolution and failure mechanisms of lithium-deficient and lithium-rich regions, both of which have been rarely achieved in previous works on the surface modification strategies for SSEs.

Consequently, we believe that this study provides novel technologies for enhancing the performance of solid-state lithium-metal batteries and new insights into their functional behavior and failure

mechanisms, showing universal feasibility for a series of other battery types as well.

Lastly, we would like to again express our sincere gratitude for your time in reviewing this article and the effort in helping us improve the quality of this work to better match the standard of *Nature Communications*.

Response to Reviewer 2

Reviewer 2 (Revision marked with purple in the MS)

Comment: *In my opinion it is a well-written and informative paper, in which the Authors proved that by the developed facile interface engineering it is possible to significantly enhance the Li/Li₇La₃Zr₂O₁₂ interface, making it suitable for the all solid-state lithium-metal batteries (ASSLMBs). There is suitable novelty present, as well as the conclusions are of practical importance. The proposed treatment of the garnet surface with trifluoromethanesulfonic acid resolves the most important issues associated with presence of the surface Li₂CO₃, creating at the same time beneficial LiCF₃SO₃ and LiF phases, forming a lithiophilic layer. This, in turn, enables preparing well-attached Li layers, even as thin as 0.78 μm. Importantly, the Authors proved validity of their approach in the manufactured ASSLMBs having different N/P ratio. The degradation mechanism of those cells was studied, and explained in details. There are numerous experimental data provided and appropriately commented. I think that the conclusions are fully-supported, and are of interests for the readers. Consequently, I believe that the submission can be accepted for publication, with only minor comments, as written below.*

- 1) *Regarding the TfOH-LLZTO interface (Fig. 1), top-view SEM/EDS data would be of interest, as homogeneous distribution of F and S elements could be better seen.*
- 2) *The Authors are asked to provide more precise evaluation of thickness of the Li layers (Fig. 4), as e.g. no error values are given. Also, there are some features/unevenness visible from the top-side view in Fig. 4, which for the thin layers may be somewhat problematic. This is not commented.*
- 3) *It is not clear how the EIS/DRT data were treated to prepare Figs. S6b and d. For example, how many spectra were recorded?*
- 4) *The same energy units could be provided in Fig. S17 and Note S2.*
- 5) *Ionic liquid was(?) used in the full cell assembly procedure (Fig. 5a). This should be explained in details. Microstructural characterization of the cathode side of the full cells could be also shown.*
- 6) *Microstructural data of the anode/electrolyte interface after electrochemical tests should be added, if possible.*

Comment 1. *Regarding the TfOH-LLZTO interface (Fig. 1), top-view SEM/EDS data would be of*

interest, as homogeneous distribution of F and S elements could be better seen.

Response: We sincerely thank the reviewer for the very positive recommendations and the constructive comments on our manuscript, which are helpful for further improving the quality of our work. According to the suggestion, we have included the top-view SEM and EDS images of TfOH-LLZTO to better show the distribution of F and S elements. It shows that F and S elements are uniformly distributed on the surface of LLZTO. We have made corresponding discussions and revisions to the revised Manuscript and Supplementary Information, as follows:

On Page 6 in the revised Manuscript:

“Scanning electron microscopy (SEM) images of TfOH-LLZTO reveal the uniform coverage of a modification layer on the LLZTO surface (Figure S4g). The corresponding elemental mappings obtained through energy dispersive spectroscopy (EDS) reveal homogeneously distributed F and S elements with high intensity (Figure S4h, i), evidencing the successful formation of F- and S-containing modification layer on the surface of TfOH-LLZTO.”

On Page 8 in the revised Supplementary Information:

Figure S4. (a) Top-SEM, (b) EDS of elemental F, S, La, and Zr, and (c) EDS spectra of LLZTO exposed to air for 1 h followed by TfoH surface treatment. (d) Top-SEM, (e) EDS of elemental F, S, La, and Zr, and (f) EDS spectra of LLZTO exposed to air for 12 h followed by TfoH surface treatment. (g) Top-SEM, (h) EDS of elemental F, S, La, and Zr, and (i) EDS spectra of LLZTO exposed to air for 24 h followed by TfoH surface treatment.

Comment 2. The Authors are asked to provide more precise evaluation of thickness of the Li layers (Fig. 4), as e.g. no error values are given. Also, there are some features/unevenness visible from the top-side view in Fig. 4, which for the thin layers may be somewhat problematic. This is not commented.

Response: Thank you for this valuable suggestion. In the revised version, we have selected five random regions and measured the thickness of the Li layer. In addition, an error bar chart was included for the thickness of the Li layer. In addition, we successfully constructed an ultra-thin lithium metal layer on a larger piece of LLZTO (1.43 cm in diameter) and observed the morphology characteristics of the lithium metal layer from both cross-section and top-section view, which showed similar morphology as the one with the smaller size (1.2 cm in diameter). It demonstrates this strategy's superior stability and generalizability for the thickness control of ultra-thin lithium anodes.

We have made corresponding revisions to the revised Manuscript and Supplementary Information, as follows:

On Page 16 of the revised Manuscript:

“...The thickness assessment and corresponding standard deviation are depicted in Figure S11. Additionally, to further confirm the consistency of this technology for fabricating ultra-thin lithium layers, a large LLZTO (1.43 cm in diameter) was adopted and coated with ultra-thin lithium metal layers of 0.78 μm and 7.54 μm thick, as depicted in Figure S12a, b. The morphology characterizations of the lithium metal layer from cross-view SEM images show a similar morphology to that of the smaller size (1.2 cm in diameter). In the top-view SEM images, the surfaces of these ultra-thin lithium layers are smooth and uniform (Figure S12c, d). It demonstrates the superior stability and generalizability of the thickness controllable preparation strategy for ultra-thin lithium anodes.”

On Page 26 of the revised Manuscript:

“...Image J is used to measure the thickness of the lithium layer in the image.⁷³”

On Page 32 of the Revised Manuscript:

“73. Schindelin J, et al. Fiji: an open-source platform for biological-image analysis. Nat Methods 9, 676-682 (2012).”

On Page 15-16 in the revised Supplementary Information:

Figure S11. The thickness of the lithium metal anodes in Figures 4b-g. Error bars are s.d.; $n = 5$.

Figure S12. Cross-view SEM images of interfaces composed of LLZTO and lithium metals of (a) 0.78 μm and (b) 7.54 μm . Cross-view SEM images of interfaces composed of LLZTO and lithium metals of (c) 0.78 μm and (d) 7.54 μm .

Comment 3. It is not clear how the EIS/DRT data were treated to prepare Figs. S6b and d. For example, how many spectra were recorded?

Response: Thank you for this valuable comment. For preparing Fig. 3b-d and Fig. S8d, seven sets of EIS/DRT data were collected, respectively. For preparing Fig. S8b, three sets of EIS/DRT data were collected. In the revised version, we have added the amount of EIS and corresponding DRT-converted data used in the figures. We have made corresponding revisions to the revised Manuscript and Supplementary Information, as follows:

On Page 15 in the revised Manuscript:

“b The DRT transition result of EIS recorded in Figure a at the current density of 0.3 mA cm⁻². Seven sets of data were selected from the original data for plotting.”

“c, d Evolution of polarization voltage and DRT transition result of EIS recorded during unidirectional charging in a Li/TfOH-LLZTO/Li cell at 0.5 mA cm⁻². Seven sets of data were selected from the original data for plotting in Figure d.”

On Page 12 in the revised Supplementary Information:

“.....(b) The DRT transition result of EIS was recorded in (a) at the current density of 0.3 mA cm⁻². In this figure, three sets of data are recorded before the cycle, after the first cycle, and after the second cycle. (c) Evolution of polarization voltage and (d) DRT transition result of EIS recorded during unidirectional charging in Li/air-LLZTO/Li cell at 0.5 mA cm⁻². In Figure S8d, seven sets of data are recorded by the step of stripping 1 mAh cm⁻².”

Comment 4. *The same energy units could be provided in Fig. S17 and Note S2.*

Response: We have unified the unit of Gibbs free energy into kJ mol⁻¹. We have made corresponding revisions to the revised Supplementary Information as follows:

On Page 4 in the revised Supplementary Information:

“2LiCF₃SO₃+16Li=3Li₂O+6LiF+Li₂S+Li₂SO₃+Li₂C₂ ΔG=-13.5 kJ mol⁻¹ T=300 °C (Equation S3)”

On Page 27 in the revised Supplementary Information:

Figure S23. Changes in Gibbs free energy of the reaction Equation S4.

On Page 29 in the revised Supplementary Information:

Figure S25. Changes in Gibbs free energy of the reaction Equation S5.

Comment 5. *Ionic liquid was(?) used in the full cell assembly procedure (Fig. 5a). This should be explained in details. Microstructural characterization of the cathode side of the full cells could be also shown.*

Response: Thank you for this suggestion. The ionic liquid used in the cathode side is lithium hexafluorophosphate liquid electrolyte. The purpose of using this ionic liquid is to improve the contact between the cathode and LLZTO. Considering the fact that liquid ionic electrolyte was added on the

cathode side to improve the interfacial contact between the SSE and cathode, we further conducted additional experiments on constructing all-solid-state batteries using polymerized solid electrolyte at the cathode side to improve the interfacial contact between the cathode and LLZTO. This battery has been cycled stably for 28 cycles before failure. This demonstrates that our ultra-thin lithium process can be applied in all-solid-state batteries without any liquid components, and in the meantime, indicates that the cathode/solid-state electrolyte interfacial contact limits the battery performance in this scenario, which, however, is not the research focus in this study.

As for the microstructural characterization of the cathode side of the full cells, the addition of liquid electrolytes will affect the ion transport and chemical composition of the cathode. To avoid this interference, therefore, we added FIB-SEM and XPS characterization of the cathode to observe the changes in the microstructure and surface chemical bonding environment before and after cycling. The results showed that the cathode remained tightly bonded before and after cycling, and a fluorinated cathode-electrolyte interface layer was detected on the surface of the cathode, which is also helpful for revealing the attenuation of the cathode in SSBs as well.

On the basis of the above discussions, we have made corresponding revisions to the revised Manuscript and Supplementary Information, as follows:

On Page 17-18 in the revised Manuscript:

“A small amount of lithium hexafluorophosphate liquid electrolyte was applied only at the cathode side as a wettability additive for facilitating the infiltration of Li ions inside the cathode.”

On Page 19 in the revised Manuscript:

“...In addition, to further verify the applicational feasibility of this technology for SSLMBs, we assembled SSLMBs with solid polymer electrolytes (SPEs) composed of succinonitrile (SN) and lithium bis(trifluoromethanesulfonyl)imide (LiTFSI) between TfoH-LLZTO and cathode to improve the interfacial contact (Figure S17). As shown in Figure S18, the assembled battery can be stably cycled for 28 cycles at 0.5 C, with a capacity retention of 84%, which is much higher than that of SSLMBs using the SPEs on both sides (39%). This again demonstrated the superiority of TfoH-LLZTO in

improving battery performance by enhancing contact at the SSE/Li interface.”

On Page 20 in the revised Manuscript:

On Page 23 in the revised Manuscript:

“Additionally, the microstructure characteristics of NCM cathode materials in SSLMBs were investigated using focused ion beam-scanning electron microscopy (FIB-SEM) images and EDS. As shown in Figure S28, the cathode materials exhibit tight and uniform contact between the components of the cathode materials, and there are no cracks inside the cathode particles before or after cycling. These phenomena indicate that cathode materials have superior ion transport performance in SSLMBs. Next, XPS measurements revealed the chemical bonding environment of the cathode-electrolyte interface (CEI) formed on the NCM cathode before and after cycling (Figure S29). In the O 1s spectrum of NCM particles after cycling, peaks belonging to C=O (533.12 eV), C-O (534.34 eV), and P-O (531.26 eV) can be observed (Figure S29a). Peaks of Li_xPO_yF_z (687.67 eV), C-F (689.10 eV), and LiF (686.01 eV) were observed in the F 1s spectrum (Figure S29b). In the P 2p spectrum, two phosphorus peaks, Li_xPO_yF_z (133.33 eV) and Li_xPF_y (134.86 eV), originating from unexpected ionic

conductors, were fitted (Figure S29c). It indicates that the added lithium hexafluorophosphate liquid electrolyte forms a CEI containing F element on the surface of NCM, and at the same time, it undergoes certain decomposition after cycling to generate unexpected $\text{Li}_x\text{PO}_y\text{F}_x$ and Li_xPF_y . This decomposition is not conducive to the cycling of the batteries, which may be the reason for the decay of the NCM cathodes.”

On Page 32 in the revised Manuscript:

“70. Li F, Liu Z, Liao C, Xu X, Zhu M, Liu J. Gradient Boracic Polyanion Doping-Derived Surface Lattice Modulation of High-Voltage Ni-Rich Layered Cathodes for High-Energy-Density Li-Ion Batteries. *ACS Energy Lett* **8**, 4903-4914 (2023).”

On Page 21-22 in the revised Supplementary Information:

Figure S17. Optical photos of the SPE on the surface of LLZTO before and after polymerization.

Figure S18. Cycling performances of NCM/SPE/TfOH-LLZTO/Li and NCM/SPE/Li cells at 50 °C. Voltage profiles of (b) NCM/SPE/TfOH-LLZTO/Li and (c) NCM/SPE/Li cell at 50 °C.

On Page 32 in the revised Supplementary Information:

Figure S28. Cross-view FIB-SEM and EDS images of the NCM electrode (a, b) before cycling and (c, d) after cycling.

On Page 33 in the revised Supplementary Manuscript:

Figure S29. (a) XPS spectra of O 1s element for the NCM before and after cycling. XPS spectra of (b) F 1s and (c) P 2p elements for the NCM after cycling.

Comment 6. Microstructural data of the anode/electrolyte interface after electrochemical tests should be added, if possible.

Response: Thank you for your suggestion. In the revised version, we have added microstructural data on the TfOH-LLZTO/Li interface after electrochemical testing. We found that Li-7.54 still maintains close contact with TfOH-LLZTO after cycling, and the surface is flat and smooth, indicating its superiority in the cycling process. We have made corresponding revisions to the revised Manuscript and Supplementary Information as follows:

On Page 22 in the revised Manuscript:

“Additionally, the morphological characteristics of lithium metal anodes post-cycling were investigated. Following cycling, Li-0.78, while tightly adherent to LLZTO (Figure S26a), exhibited irregular surface protrusions (Figure S26b) and increased surface roughness compared to its pre-cycling state (Figure S12). In contrast, post-cycling, Li-7.54 maintained a relatively smooth surface and demonstrated robust bonding with LLZTO (Figure S27). This underscores the advantageous cycling performance of Li-7.54, conducive to the uniform deposition of lithium ions.”

On Page 29-30 in the revised Supplementary Information:

Figure S26. (a) Cross-view SEM and (b) Top-view SEM images of interfaces composed of LLZTO and lithium metals of 0.78 μm after cycling.

Figure S27. (a) Cross-view SEM and (b) Top-view SEM images of interfaces composed of LLZTO and lithium metals of 7.54 μm after cycling.

Lastly, we would like to express our sincere gratitude again to the reviewers for their time reviewing this article and their efforts in helping us improve the quality of this work to better match the standards of *Nature Communications*.

REVIEWERS' COMMENTS

Reviewer #1 (Remarks to the Author):

Different from reported works, this manuscript highlights the controllable thickness of lithium anodes by regulating air exposure and surface treatment to form a super lithiophilic layer on LLZTO. Importantly, the authors propose multi-dimensional compositional evolution and failure mechanisms for the lithium-deficient and -rich regions of lithium anodes, which is of broad interest for the readers. In addition, after careful revision, more data and analysis have been provided to support the conclusions and improve the quality of this manuscript. Therefore, this manuscript can be accepted for publication in this journal.

Reviewer #2 (Remarks to the Author):

I don't have any further comments. I believe that the paper can be accepted as is.